# The digital terrain model in the computational modelling of the flow over the Perdigão site: the appropriate grid size

José Manuel Laginha Mestre da Palma[1], Carlos Alberto Madureira da Silva[1], Vitor Manuel Martins Gonçalves da Costa Gomes [1], Alexandre Costa da Silva Lopes[1], Teresa Simões Esteves[2], Paula Alexandra da Silva Costa[2], and Vasco Teófilo Preto Batista[1]

[1]Faculty of Engineering of the University of Porto (FEUP), Mechanical Engineering Department, Rua Dr. Roberto Frias, s/n, 4200-465 Porto, Portugal
[2]National Laboratory of Energy and Geology (LNEG), Estrada da Portela, Bairro do Zambujal, Apartado 7586, Alfragide, 2610-999 Amadora, Portugal

**Correspondence:** José M.L.M. Palma (jpalma@fe.up.pt)

**Abstract.** The digital terrain model (DTM), the representation of Earth's surface at regularly spaced intervals, is the first input in the computational modelling of atmospheric flows. The ability of computational meshes based on high (2 m, airborne laser scanning), medium (10 m, military maps) and low (30 m, shuttle mission, SRTM) resolution DTMs to replicate the Perdigão experiment site was appraised in two ways: by their ability to replicate the two main terrain attributes, elevation and slope, and by their effect on the wind flow computational results. The effect on the flow modelling was evaluated by comparing the wind speed, wind direction and turbulent kinetic energy by VENTOS®/2 at three locations, representative of the wind flow in the region. It was found that the SRTM was not an accurate representation of the Perdigão site. A 40 m mesh based on the highest resolution data, yielded at five reference points an elevation error of less than 1.4 m and an RMSE of less than 2.5 m, compared to 5.0 m, in the case of military maps and 7.6 m in the case of SRTM. Mesh refinement beyond 40 m yielded no or insignificant changes on the flow field variables, wind speed, wind direction and turbulent kinetic energy. At least 40 m horizontal resolution–*threshold resolution*–, and based on topography available from aerial survey, is recommended in computational modelling of the flow over Perdigão.

# 1 Introduction

A Digital Terrain or Elevation Model (DTM or DEM) is a representation of the Earth's surface elevation at regularly spaced horizontal intervals. Although the terrain model is the first input in computational modelling of atmospheric flows, its impact on flow results has not been a matter of concern, because the spatial resolution of publicly available DTMs is higher than the size of the computational grid often used to resolve the terrain. However, before a fluid flow database (Mann et al., 2017) can be used as a reference in flow model appraisal and development, the impact of the terrain modelling must be assessed. For studies of the atmospheric flow over Perdigão the publicly available DTMs were considered not accurate enough (Mukherjee et al., 2013; Simpson et al., 2015) and an airborne laser scanning campaign of the region was carried out in 2015; first to assist the design of the Perdigão campaigns in 2015 and 2017 (cf., Vasiljević et al., 2017; Fernando et al., 2019) and second, to provide the high resolution terrain data for computational flow modelling, on par with the resolution provided by the large number of measuring equipment within a small region.

The Perdigão site is located in the municipality of Vila Velha de Ródão, in the centre of Portugal (608250E, 4396621N: ED50 UTM 29N or in WGS84, $39°42'38.5''$N $7°44'18.5''$W). It is comprised by two parallel ridges, about 500 m elevation, 4 km length, SE-NW orientation, and distanced around 1.4 km from each other. The land is covered by a mixture of farming areas and patches of eucalyptus (*Eucalyptus globulus*) and pine trees (*Pinus pinaster*). The dominant winds are perpendicular to the ridges, assuring a largely two-dimensional flow.

The accuracy of a DTM depends on the data collection techniques, data sampling density and data post-processing, such as grid resolution and interpolation algorithms. In computational modelling of atmospheric flows, DTMs are often used from photogrammetry or satellite interferometry, such as *SRTM, Shuttle Radar Topography Mission*, (Farr et al., 2007) or *ASTER, Advanced Spaceborne Thermal Emission and Reflection Radiometer* (Yamaguchi et al., 1998), freely available at https://earthexplorer.usgs.gov/. SRTM has the widest cover and is the most commonly used terrain data. Its latest version (V3.0 1", 2014) is 1 arc-second on most of the planet's surface, i.e. about 23.75 m resolution and an absolute height error equal to 6.2 m at Perdigão's latitude (Farr et al., 2007). With the advent of high resolution techniques such as lidar aerial survey, terrain data has become available with resolutions above 10 m and vertical accuracy typically below 0.2 m (Hawker et al., 2018), and the question is whether such high resolution is needed in the computational modelling of atmospheric flows over complex terrain.

## 1.1 Literature review

Grid independent calculations is a concept very dear to computational fluid dynamics practitioners (e.g., Roache, 1998), as a mean for reducing discretisation errors. However, its application in the context of atmospheric flows is not that simple, because every level of grid refinement brings another level of surface detail; see for instance the coastline paradox (Mandelbrot, 1967, 1982). In this case, because the flow is driven by topography, the flow model results are directly correlated to the terrain data and our problem is common to what can be encountered in geomorphology, with applications in hydrology (e.g., Zhang and Montgomery, 1994; Wise, 2000; Deng et al., 2007; Savage et al., 2016), where the DTM grid size affects the drainage area. In

spite of its importance, to our knowledge, there is no systematic study on the appropriate grid size for resolving the terrain in microscale modelling of atmospheric flow over complex terrain.

Work has been done in quantifying the impact of using different DTMs and resolution on terrain attributes, such as elevation, slope, plan and profile curvature, and topographic wetness index. For instance, Mahalingam and Olsen (2016) notes that DEMs are often obtained and resampled without considering the influence of its source and data collection method. Finer meshes do not necessarily mean higher accuracy in prediction (with examples for landslide mapping where terrain slope has a great influence) with the DEM source being an important consideration.

DeWitt et al. (2015) compared several DEMs (USGS, SRTM, a statewide photogrammetric DEM and ASTER) to a high-accuracy lidar DEM to assess their differences in rugged topography through elevation, basic descriptive statistics and histograms. Root mean square error ranged from 3 (using photogrammetric DEM) to circa 15 (using SRTM) or 17 m (using ASTER).

Deng et al. (2007) indicated that the mesh resolution can change not only terrain attributes in specific points but also the topographic meaning of attributes at each point. They concluded that variation of terrain attributes were consistent with resolution change and that the response patterns were dependent on the landform classes of the area. Deng et al. (2007) introduced the concept of threshold resolution, i.e. the resolution beyond which the model quality deteriorated quickly, but below which no significant improvement in modelling results was observed.

Florinsky and Kuryakova (2000) developed an experimental three-step statistical method to determine an adequate resolution in DEM to represent topographic variables and landscape properties at a micro-scale (exemplified by soil moisture) by performing a set of correlation analysis between resolutions.

Diebold et al. (2013) showed the effect of grid size in LES of flow over Bolund. Lange et al. (2017) addressed the question of how to represent the small topographic features of Bolund in wind tunnel modelling, comparing a round and a sharp edge of a cliff in a wind tunnel, to conclude that the cliff with the sharp edge gives an annual energy production of a wind turbine near the escarpment that is $20\,\%$ to $51\,\%$ of the round-edge case.

## 1.2 Objectives and outline

The objective of the present study is to determine the terrain resolution required to accurately resolve the atmospheric flow over Perdigão and mountainous terrain in general. One needs to assess the terrain horizontal resolution before assessing the effect of other (also important) causes of differences between experimental and computational results. Many computational studies based on Perdigão data are expected and it is important to asses the terrain resolution requirements first.

In what follows, we describe the techniques used for aerial and terrestrial surveying (section 2), plus the post-processing of those data and the determination of the main geometrical parameters of the Perdigão site (section 3). The results on terrain attributes and computational flow modelling are the subject of sections 4 and 5. The paper ends (section 6) with conclusions and recommendations on the most appropriate DTM and grid size required in the computational modelling of the flow over Perdigão.

## 2 Topographical surveying: equipment and techniques

### 2.1 Airborne laser scanning (2015)

The lidar aerial survey (Mallet and Bretar, 2009) was performed on 15 March 2015 by NIRAS (2015), with assistance from Blom TopEye. The survey covered an area equal to $22\,071\,075\,\mathrm{m}^2$ (Figure 1) and was completed in one session, at an altitude of 500 m with a TopEye system S/N 444 and a camera mounted on a helicopter. The number of points of the lidar point cloud was approximately 993198375, an average point density inside the project area equal to 45 points/m$^2$ and 12.6 points/m$^2$, if restricted to the *ground* class points (Figure 1a and section 3.1). The photography (a total of 744 photos, stored as $300\,\mathrm{m}\times300\,\mathrm{m}$ tiles) was performed with a Phase One iXA180 camera, a medium format, 10328 pixel $\times$ 7760 pixel sensor resolution, yielding a ground resolution equal to $4.7\,\mathrm{cm}$ (Figure 1b).

Lidar data was checked for point density control by Blom's software TPDS (TopEye Point Density and Statistics), with the area being fully covered by lidar data with exceptions for watersheds. GPS signal was processed using data from three Portuguese reference network stations (CBRA, MELR and PORT, cf. DGT (2017)), after assistance by the Portuguese National Mapping Agency (*Direção-Geral do Território, Divisão de Geodesia*). Discrepancies between flight lines (based on Blom's software TASQ, TopEye Area Statistics and Qualities, calculated on subareas of 1 m and after matching of 204104275 observations) showed a maximum altitude deviation and RMSE equal to 0.490 m and 0.061 m. In $75\,\%$ of the subareas the RMSE was lower than $60\,\mathrm{mm}$.

The raw data of the NIRAS (2015) campaign comprised the lidar point cloud in LASer (LAS) (version 1.2) format and the ortophotos in 20 and 5 cm resolution; for more information and availability on these data see Palma et al. (2020). The production of the digital terrain model based on the lidar point cloud is the subject of section 3.1.

### 2.2 Terrestrial surveying (2017 and 2018)

During the installation of scientific equipment (Nov 2016–May 2017), terrain elevation was measured in situ (Palma et al., 2018) for an accurate and final determination of the elevation data of part of the instrumentation. The measuring equipment was a Leica system, comprised of the following units: (1) Leica MultiStation MS50, (2) Leica Viva GS14 - GNSS Smart Antenna, (3) Leica CRT16 Bluetooth Cap and (4) Leica GRZ121 360 Reflector PRO Surveying Prism.

In 2017, a piece of land required changes for installation of tower 20/tse04. The topographic survey of that region was carried out (Alves, 2018) and incorporated in the lidar based terrain model of March 2015. This survey was performed by Spectra Physics (SP60 GNNS receiver and data collector T41 with Survey Pro software) equipment and software by Sierrasoft (PROST Premium/Topko Standart, Version 14.3).

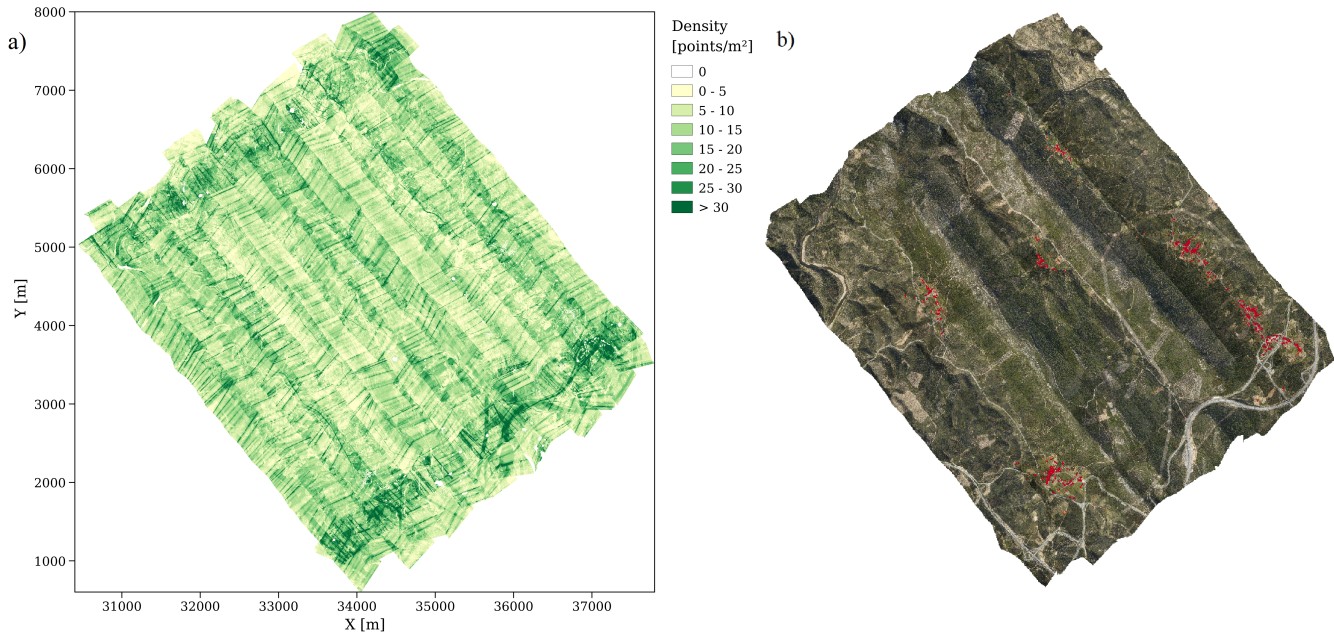

**Figure 1.** Lidar aerial survey and ortophoto: a) *ground* class points distribution; b) ortophoto (houses in red).

## 3 Terrain model

### 3.1 Lidar point cloud processing

The lidar data was classified in four data type classes (*ground*, *vegetation*, *unassigned* and *noise*) and stored in LAS File Format, and then post-processed with tools pertaining to the LAStools© software suite (LAStools, 2019) in three stages, Figure 2.

Stage 1 was concerned with the extraction of the lidar raw data. The *ground* class point cloud had irregular spacing (Figure 1a), with lower point density in regions of vegetation clumping or non-overlapping scans. Some small, distinctive areas were found to be devoid of *ground* points, due to watersheds and lidar reading or classification errors.

Stage 2 involved the reclassification of abnormal data. A first procedure was used to reclassify a particular area of *noise*-classified points into the *ground* or *vegetation* classes, which would otherwise be void of *ground* points. Points with excessive (>700 m) or negative elevations a.s.l. (above sea level) were removed during this stage. Isolated points above or below the more spatially dense point cloud, classified as *ground* or *vegetation*, were identified and removed using the `lasnoise` tool (LAStools, 2019).

In Stage 3, a Triangulated Irregular Network (TIN), based on the Delaunay triangulation, was obtained for the *ground* classified points. The DTM was obtained by interpolating the heights into a regular mesh with a resolution of $2\,\mathrm{m} \times 2\,\mathrm{m}$, the highest horizontal resolution of the terrain elevation within the Perdigão site.

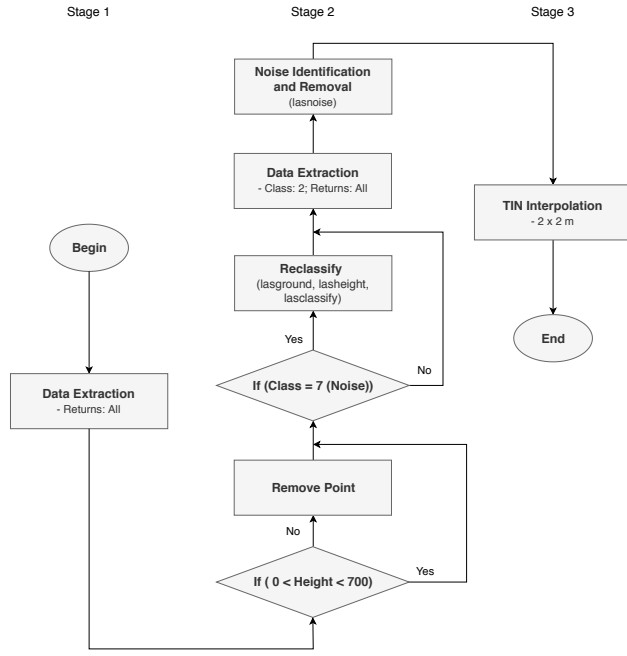

**Figure 2.** Workflow diagram for producing the terrain map using LAStools©.

### 3.1.1 Buildings

It is not clear whether a DTM should comprise buildings and other human-made artefacts that are usually part of digital surface models (DSM). In the context of the present study, buildings are long-standing structures, as a terrain feature, and we saw no reason for buildings not being part of the DTM. The houses, in Figure 1b, are family houses of about $15\,\mathrm{m} \times 15\,\mathrm{m}$ and $5\,\mathrm{m}$ height that will show on the finest mesh only and as a point elevation. Unless there are a few neighbouring buildings, the ability to resolve isolated houses is limited.

The first task, to include the building data in the DTM, embraced the digitalization of all buildings from the orthophotos. This process was needed to retain the building polygons as close as possible to their exact shape and location. The second task involved the extraction of *unassigned* data points, which included buildings and adjacent vegetation, among other structures that fell within the polygons. These points were reclassified –using the `lasclassify` tool (LAStools, 2019), to further remove adjacent and overhanging vegetation– and the resulting *building* class points were converted to the *ground* class. The third and final stage comprised the generation of a TIN from the new *ground* point cloud, followed by interpolation (`blast2dem` tool, LAStools (2019)) of the heights to a regular mesh with a resolution of $2\,\mathrm{m} \times 2\,\mathrm{m}$.

Calculations including the building data showed minor or no visible flow changes and were discarded. Nevertheless, for future use, two DTM versions, with and without buildings, are made available (Palma et al., 2020).

## 3.2 Two-dimensionality and main geometrical parameters

One of the reasons why Perdigão was selected was its geometry; namely, the parallelism between the two ridges and their large length relatively to the width, bringing the orography close to two-dimensionality.

### 3.2.1 Area of interest (AOI), reference lines and locations

For scaling and dimensional analysis, the main geometrical parameters of the Perdigão site were determined and the area of interest (AOI) was defined (Figure 3 and Table 1): a rectangular shape of approximately ($3\,\mathrm{km} \times 4\,\mathrm{km}$), with lower left corner at 608589E; 4394131N and aligned with the centreline ($\ell_C$, SE-NW direction, $135°$). This area, centred near station 131, included the SW and the NE ridge, the valley and the location of most of the instrumentation deployed in Perdigão. Note that the coordinate system was converted from ETRS89 PT-TM06 (original source) to ED50 UTM29 and will be used throughout the document as Eastings and Northings. Stations number (#) and code as in Perdigão web site (perdigao.fe.up.pt/).

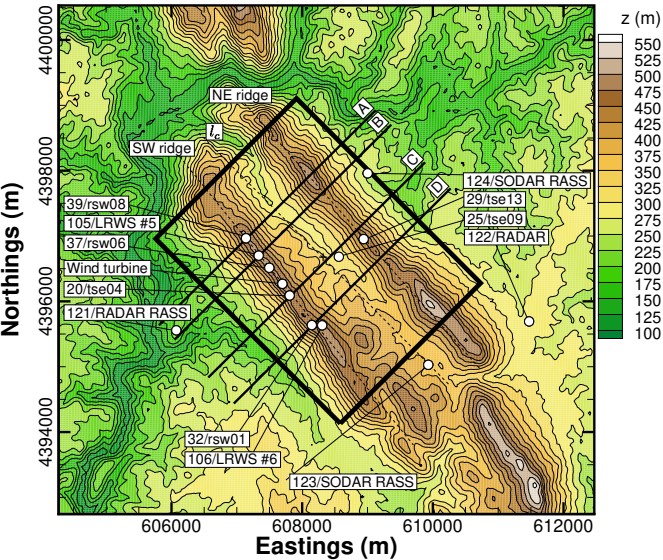

**Figure 3.** Area of interest and transects (ED50 UTM 29N).

### 3.2.2 Terrain profile and slope

The terrain profile (Figure 4) is not uniform along the valley, which becomes narrower and deeper along the SE-NW direction. For instance, the NW ridge height relative to a reference height ($h_{ref}$, the mean height of the surrounding terrain in a $20\,\mathrm{km} \times 20\,\mathrm{km}$ area) equal to 250 m a.s.l. varies between 201.1 and 251.4 m, and the distance between ridges is 1358.0 and 1480.0 m on transects A and D (Table 2).

**Table 1.** Reference points as in Figure 3 (ED50 UTM 29N).

| # | Type/Code | Eastings (m) | Northings (m) | Elevation (m) |
|---|---|---|---|---|
| | Wind turbine | 607697 | 4396268 | 484.0 |
| 20 | 20/tse04 | 607808 | 4396090 | 473.0 |
| 25 | 25/tse09 | 608561 | 4396683 | 305.3 |
| 29 | 29/rsw01 | 608939 | 4396953 | 452.9 |
| 32 | 32/rsw01 | 608149 | 4395638 | 472.1 |
| 37 | 37/rsw06 | 607498 | 4396514 | 482.5 |
| 39 | 39/rsw08 | 607140 | 4396966 | 488.9 |
| 105 | LRWS #5 | 607335 | 4396701 | 485.9 |
| 106 | LRWS #6 | 608307 | 4395634 | 486.3 |
| 121 | RADAR/RASS | 606074 | 4395558 | 223.7 |
| 122 | RADAR | 611474 | 4395697 | 288.6 |
| 123 | SODAR/RASS | 609931 | 4395029 | 361.9 |
| 124 | SODAR/RASS | 609003 | 4397960 | 258.4 |

Apart from $\ell_C$, six additional lines were defined: $\ell_{SW}$ and $\ell_{NE}$ along the SW and the NE ridges, and A, B, C and D, perpendicular to the ridges (SW-NE direction, 225°) and related to four main transects: A and D that delimit the northernmost (station 39) and southernmost (station 32) locations of the great majority of the instrumentation; and transects B and C that delimit a narrower region, determined by locations of stations 105/LRWS#5 and 20/tse04.

Other geometric variables (Figure 4) are the height of ridges ($h_{SW}$ and $h_{NE}$) and valley ($h_{val}$) relative to the reference height ($h_{ref}$), the half-widths of the ridges ($l_{SWw}, l_{SWe}, l_{NEw}$ and $l_{NEe}$) at half-height ($h_{SW}/2$ and $h_{NE}/2$), and the distance between ridges, $\ell$.

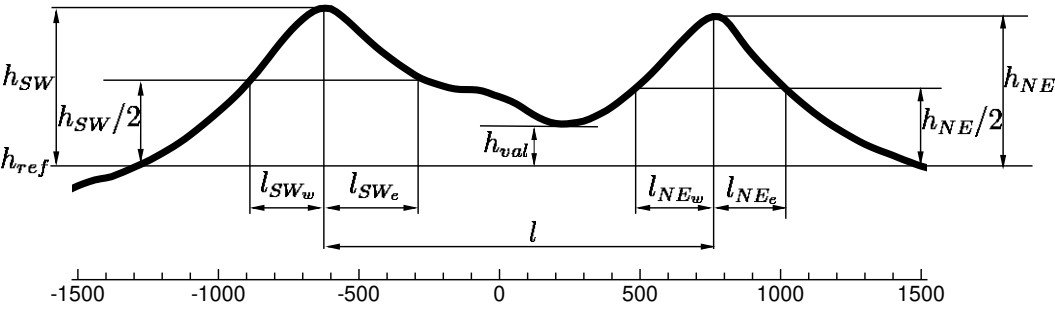

**Figure 4.** Terrain profile: geometrical parameters.

**Table 2.** Main geometric variables of transects A, B, C and D (slope (S) in degrees ($^\circ$), and length and elevation ($\ell$ and $h$) in metre, $h_{ref} = 250$ m a.s.l.).

|  | A | B | C | D | Average |
|---|---|---|---|---|---|
| $h_{SW}$ | 237.2 | 237.3 | 228.3 | 222.0 | 231.2 |
| $h_{NE}$ | 251.4 | 212.7 | 205.3 | 201.1 | 217.6 |
| $h_{val}$ | 24.5 | 31.1 | 46.8 | 65.2 | 41.9 |
| $l_{SWw}$ | 277.8 | 214.0 | 232.0 | 212.3 | 234.0 |
| $l_{SWe}$ | 270.0 | 305.0 | 402.9 | 432.0 | 352.5 |
| $l_{NEw}$ | 286.9 | 320.2 | 249.5 | 268.7 | 281.3 |
| $l_{NEe}$ | 258.4 | 245.0 | 221.9 | 261.7 | 246.7 |
| $l$ | 1358.0 | 1384.0 | 1412.0 | 1480.0 | 1408.5 |
| $S_{SWw}$ | 33.6 | 37.5 | 36.8 | 40.0 | 37.0 |
| $S_{SWe}$ | -45.1 | -29.6 | -22.7 | -21.1 | -29.6 |
| $S_{NEw}$ | 30.7 | 25.7 | 27.8 | 25.7 | 27.5 |
| $S_{NEe}$ | -35.7 | -30.1 | -30.7 | -24.1 | -30.1 |

The ridges' orientation was determined by a linear regression of two $z$ maxima, for each $j$ (mesh oriented with SW-NE direction) on a 20 m grid, between transects A and D ($\approx 1650$ m) and between transects B and C ($\approx 530$ m). The deviations from parallelism are 4.3° if restricted to the region between A and D. Between transects B and C, where the core of the instrumentation was, the ridges were parallel within 2.8°; i.e., 139.1° and 136.3° in the case of SW and NE ridges. The slope ($S = |\text{atan}(h_{SW,NE}/2)|/\ell_{SW,NE}$), also on a 20 m grid varies between 21.08° and 45.09°, always above the threshold for flow separation under neutral conditions (Wood, 1995).

## 4 Digital terrain model: results and discussion

The terrain elevation and slope are the two main terrain attributes for classification of terrain complexity and the ones to replicate accurately by terrain models. In this section, three DTMs of the Perdigão site are analysed within the AOI, by comparing terrain elevation and slope on meshes based on these terrain models with the terrain elevation and slope measured by the lidar aerial survey data within the AOI.

The three DTMs (Figure 5 and section 4.4) were the following: (1) ALS, the area sampled by lidar with a 2 m resolution; (2) Military, the Portuguese Army cartography around Perdigão, 10 m horizontal resolution available from Portuguese Army Geospatial Information Centre (*CIGeoE Centro de Informação Geoespacial do Exército*), a total of eight sheets (numbers 290.4, 291.3, 302.2, 302.4, 303.1, 303.3, 313.2 and 314.1) at a scale equal to 1:25000; and (3) SRTM, the SRTM 30 m. Information on availability of these data can be found in Palma et al. (2020) and under the heading *Data availability*, at the end of the present study.

Because the ALS was the highest resolution map, and the most accurate representation of the terrain in Perdigão, it was the one against which the accuracy of alternative terrain data sources was evaluated.

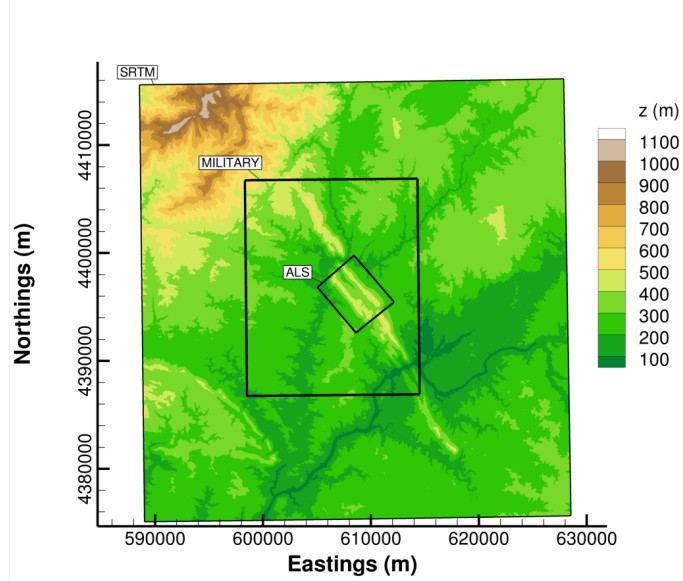

**Figure 5.** Total area comprised by SRTM, military and airborne data.

Concerning the terrestrial surveys in 2017 and 2018 (section 2.2), a sample of those measurements confirmed the high quality of lidar airbone measurements. The survey carried in 2018, showed that the terrain change due to installation of tower 20, yielded alteration of the terrain that were not significant (lower than 1 m).

### 4.1 Mesh generation

For comparison of terrain attributes, elevation and slope, regularly spaced meshes of 80, 40, 20 and 10 m (size, $n_i \times nj$, respectively $39 \times 51$, $77 \times 101$, $153 \times 201$ and $305 \times 401$) were generated within the AOI. The resampling procedure was similar to Deng et al. (2007); i.e., one out of two points was retained to assure that every point in the coarser resolutions existed on the finer ones. Coarser meshes are resampled versions of the 2 m resolution mesh, obtained by removing additional nodes.

### 4.2 Elevation at five reference points

Five points (Table 1) were selected for DTM comparison: towers 20/tse04, 25/tse09 and 29/tse13, the three 100 m meteorological towers, comprising a transect aligned with the dominant wind direction, and tower 37/rsw06 and the wind turbine location, along the SW ridge.

Figure 6 shows the absolute error ($z_{error} = z_{80,40,20,10} - z_2$), difference between the elevation at a given mesh and DTM source, with respect to the terrain elevation on the reference mesh ($ALS_2$). In the case of SRTM based meshes (Figure 6a), the

error tends to a plateau at resolutions equal to 40 m. Similar behaviour is found in the case of the Mil database (Figure 6b), but at 20 m resolution; 20 and 10 m mesh increases the error at 20/tse04 and meshes at higher resolution to the uncertainty of this database must be avoided. Contrary to the SRTM and Mil, when using ALS (Figure 6c), with mesh refinement there is a noticeable error reduction at all 5 points.

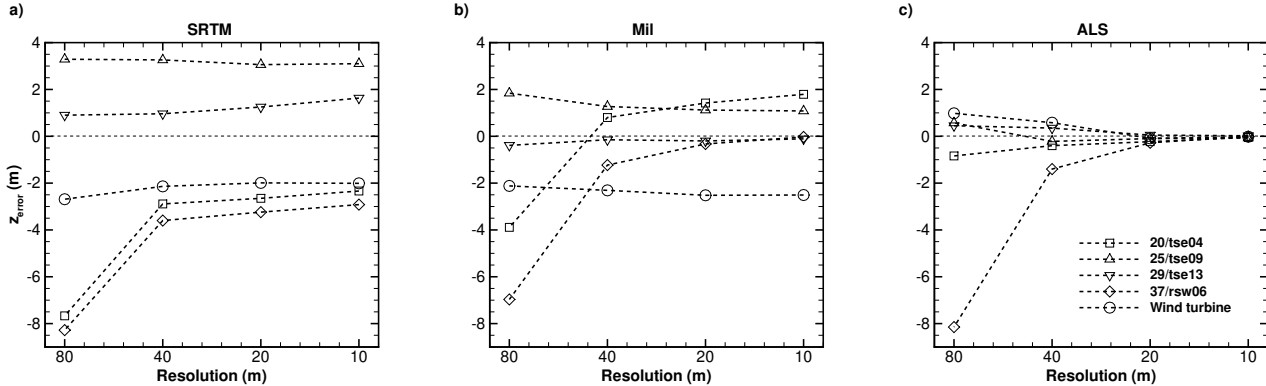

**Figure 6.** Impact of mesh resolution on reference points: a) SRTM; b) Mil; c) ALS.

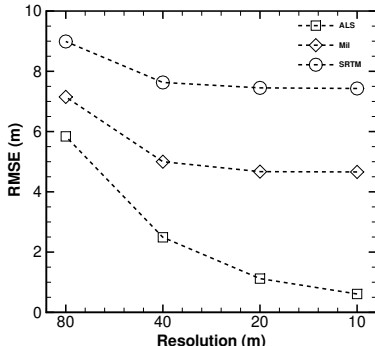

**Figure 7.** Impact of mesh resolution on RMSE.

### 4.3 Elevation and slope in the area of interest (AOI)

As the DTM quality increased from SRTM to Mil and ALS, the maximum terrain elevation ($z_{Max}$) also increased, from 530.2 to 540.5 and 540.8 m, and the minimum ($z_{Min}$) decreased from 165.0 to 158.8 and 156.8 m (Table 3). Maxima and minima terrain elevations are set by the DTM source; maxima and minima are similar for a given DTM regardless of the grid size, which was a consequence of the procedure for mesh refinement. The 10 m difference between SRTM and the ALS values is

consistent with the RMSE of SRTM, equal to 6.2 m for Eurasia (Table 1, Farr et al., 2007) and also with the conclusions by DeWitt et al. (2015).

**Table 3.** Maxima and minima terrain elevation, based on SRTM, Mil and ALS data.

| | $z_{Max}$ (m) | | | $z_{Min}$ (m) | | |
|---|---|---|---|---|---|---|
| Mesh | SRTM | Mil | ALS | SRTM | Mil | ALS |
| 80 | 530.2 | 538.4 | 537.3 | 165.0 | 159.4 | 157.0 |
| 40 | 530.2 | 538.4 | 537.9 | 165.0 | 159.4 | 157.0 |
| 20 | 531.1 | 540.0 | 539.4 | 165.0 | 158.8 | 156.8 |
| 10 | 531.4 | 540.5 | 540.8 | 165.0 | 158.8 | 156.8 |
| 2 | - | - | 541.1 | - | - | 156.8 |

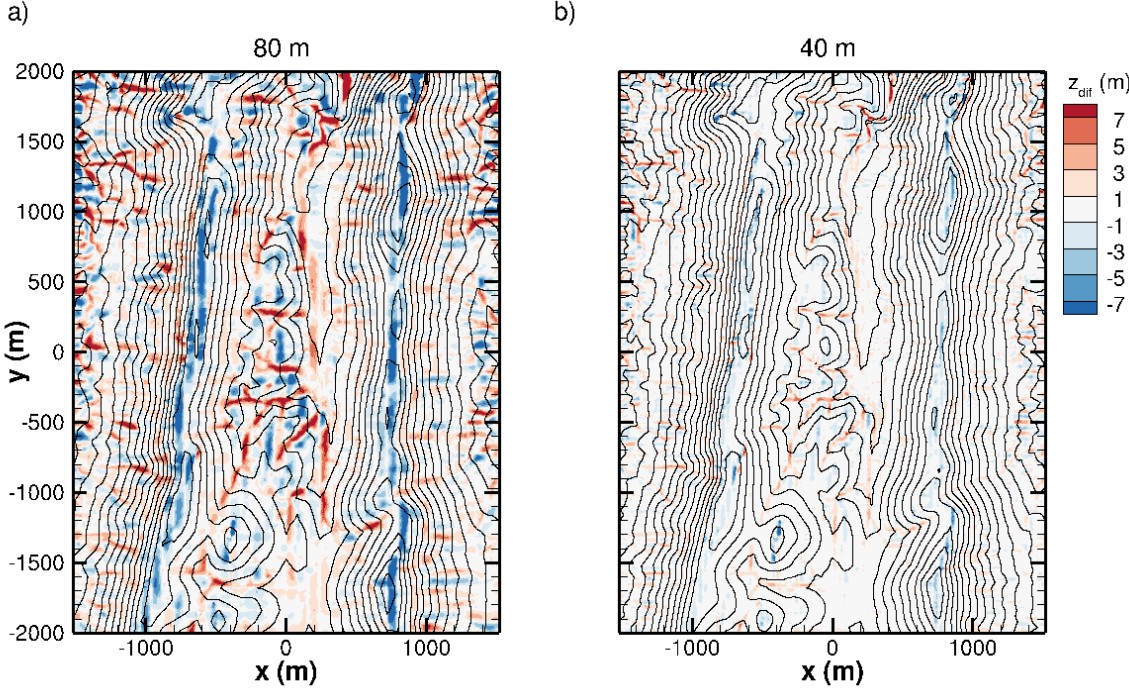

**Figure 8.** Elevation error of resampled meshes (ALS terrain data) over the AOI surface: a) 80 m mesh; b) 40 m mesh.

The error distribution (Figures 8 and 10a) shows an overprediction of the terrain elevation along the valley and an underprediction along the ridges, with both much reduced between the 80 and the 40 m resolution meshes, with the latter showing a mostly uniform error distribution of around 1 m (Figure 8b).

**Table 4.** Maxima and minima slope in the $x$ (SW-NE, 225°) direction, based on SRTM, Mil and ALS terrain data.

| | $S_{Max}$ (°) | | | $S_{Min}$ (°) | | |
|---|---|---|---|---|---|---|
| Mesh | SRTM | Mil | ALS | SRTM | Mil | ALS |
| 80 | 39.00 | 38.27 | 37.31 | -36.11 | -37.99 | -37.33 |
| 40 | 41.61 | 43.10 | 44.24 | -38.64 | -47.61 | -51.74 |
| 20 | 44.65 | 49.36 | 55.85 | -47.18 | -55.19 | -59.31 |
| 10 | 47.86 | 51.74 | 64.76 | -49.02 | -61.31 | -67.81 |
| 2 | - | - | 75.91 | - | - | -81.13 |

The RMSE error (Figure 7) over the whole AOI is consistent with the elevation error and the inherent uncertainty of every DTM source; with mesh refinement every DTM tends to its resolution level. The minimum RMSE of SRTM, Mil and ALS are 7.43, 4.66 and 0.61 m at resolutions of 10 m.

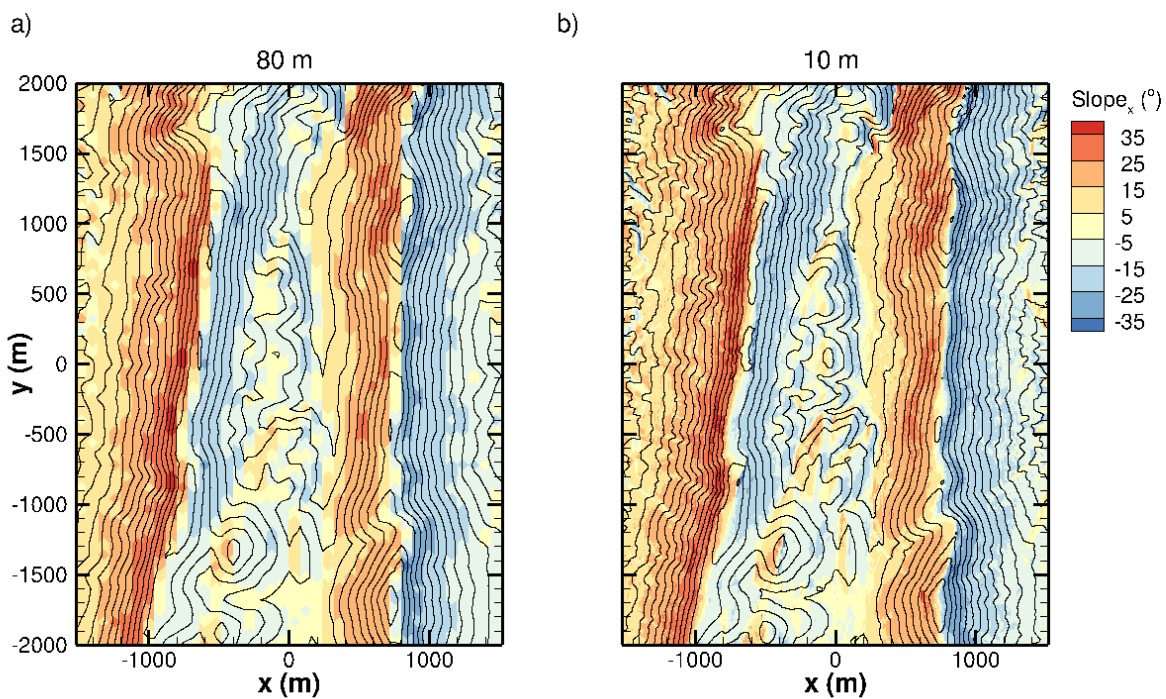

**Figure 9.** Slope in $x$ (SW-NE, 225°) direction with different resolutions mapped on AOI's surface (ALS terrain data): a) 80 m mesh; b) 10 m mesh.

The maximum slope (55.85° and 64.76°) was about $50\%$ higher on 20 and 10 m meshes compared with the coarser resolution (37.31° and 44.24° meshes, 80 and 40 m), Table 4. The negative slope increased from $-37.33°$ to $-67.81°$, as the resolution increased from 80 to 10 m. The histogram of the slope in the $x$ (SW-NE, 225°) direction (Silva, 2018) shifted to the right, as

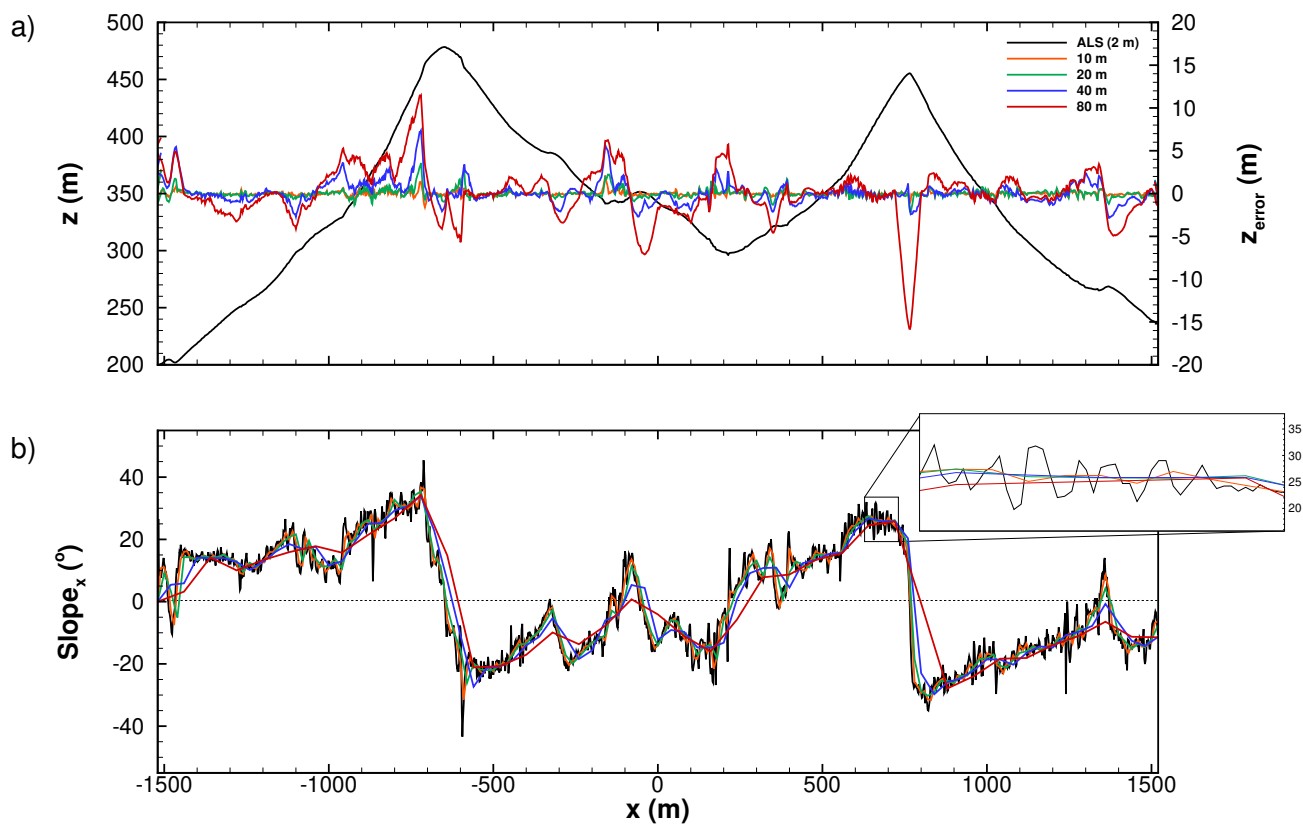

**Figure 10.** Terrain along transect C (see Figure 3), ALS based meshes: a) terrain profile and elevation error; b) slope in the $x$ direction.

the content at low slopes, decreases and more and more higher slope locations were resolved. Because the ridges are quasi two-dimensional, the $y$ (SE-NW, 135°) direction slope was residual (Silva, 2018) compared to the $x$ direction slope (Figure 9) and is not shown here.

The larger errors occurred at locations of higher slope (Figure 10) and these are the locations where the grid refinement was also the most effective in reducing the elevation error. For instance, errors equal to $11.5\,\mathrm{m}$ and $-15.8\,\mathrm{m}$ (at $x = -720\,\mathrm{m}$ and $x = 766\,\mathrm{m}$) on a 80 m mesh were reduced to $7.5\,\mathrm{m}$ and $-2.5\,\mathrm{m}$ on a 40 m mesh.

### 4.4   Spectra analysis

Spectra of terrain elevation show the ALS resolution one order of magnitude higher compared to SRTM data (Figure 11). The

figure also displays two scaling ranges, typical of global topographies (e.g., Nikora and Goring, 2004), with exponents equal to -7/4 and - 11/3.

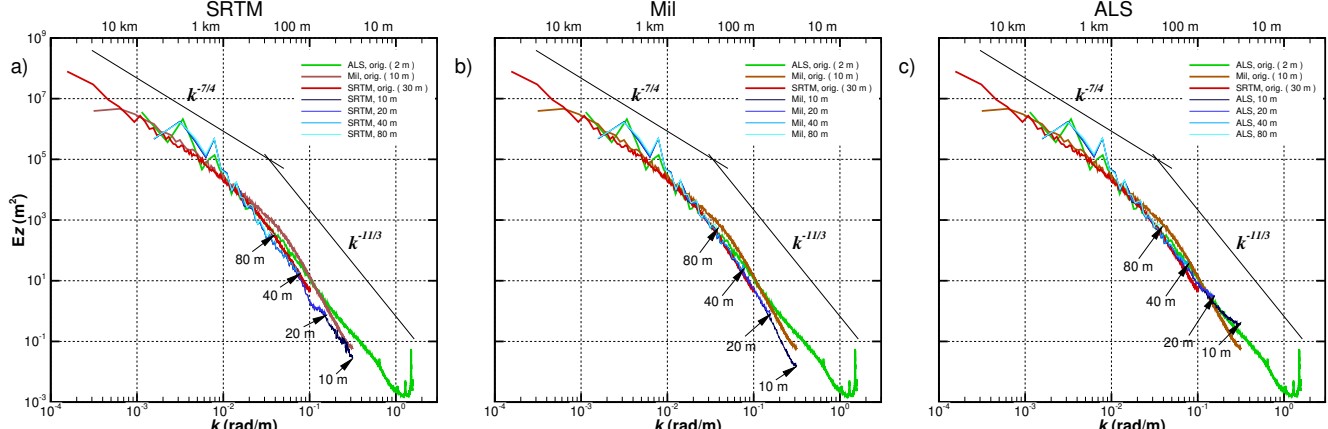

**Figure 11.** Spectra analysis for DTM and meshes: a) SRTM; b) Mil; c) ALS.

As expected, there is an increase in resolved spectral range with mesh refinement and an overlap between meshes with ALS data. In the case of SRTM and Mil based meshes (Figures 11a and 11b), linear refinements (20 and 10 m meshes) cannot replicate the decay for higher frequencies and overcome the inherent resolution of the original data. Mesh quality was as good
as the terrain data source. Only meshes based on the ALS (Figure 11c) have the ability to reproduce accurately the high-frequency range ($10^{-1}\,\mathrm{rad\,m^{-1}} < k < 1\,\mathrm{rad\,m^{-1}}$). SRTM is restricted to around 30 m resolution and meshes $20\,\mathrm{m} \times 20\,\mathrm{m}$ and $10\,\mathrm{m} \times 10\,\mathrm{m}$, with identical $z_{Max}$ and $z_{Min}$ (531 and 165 m), are unable to replicate the ALS measured values, $z_{Max}$ and $z_{Min}$, Table 3. Grid refinement cannot go beyond the inherent resolution of the DTM.

### 4.5 RIX (ruggedness index)

The RIX (ruggedness index) is one of the major parameters in WAsP (Mortensen et al., 2004). It has the goal of quantifying the terrain complexity. The operational envelope of WAsP corresponds to a RIX value of $0\,\%$. The ALS data shows a maximum of $23.7\,\%$ and an overall higher value of RIX (average $15.22\,\%$), while SRTM reaches $19.7\,\%$ (average $11.6\,\%$). Lower resolution terrain data underestimate the terrain complexity.

## 5 Flow modelling

In this section, the results of computational runs on different meshes are discussed. A set of experimental data (UCAR/NCAR-EOL, 2019) (30 minutes averaged) on 4 May 2017, 22:09–22:39 UTM is also included. This was the day and the time when the assumption of conditions of stationarity based on measurements at tower 20/tse04 were valid, according with Carvalho (2019), and the flow was non-stratified based on a bulk Richardson number ($R_B$) equal to -0.03

$$R_B = \frac{g\left(\overline{\theta}_{100} - \overline{\theta}_2\right)\Delta z}{\overline{\theta}_{100}\left[(U_{100})^2 + (V_{100})^2\right]}$$
(1)

where $\overline{\theta}_{100}$ and $\overline{\theta}_2$ are the mean potential temperature at $100$ m and $2$ m a.g.l (above the ground level), $\Delta z = 100$ m, and $U_{100}$ and $V_{100}$ are the mean horizontal components of the velocity vector also at $100$ m a.g.l. . The temperature obtained from measurement data was converted into potential temperature using the following approximation (Stull, 1988):

$$\overline{\theta} \approx \overline{T} + \left(\frac{g}{c_p}\right) z \qquad (2)$$

The data set choice was conditioned by the computational flow model being used. Because computational results do not
consider, for instance, surface cover heterogeneity, discrepancies are expected when compared with experimental data, which are included here for guidance only.

## 5.1 Computational flow model

The computational code VENTOS®/2 (cf., Castro et al., 2003; Palma et al., 2008), developed for atmospheric flows over complex terrain, was used in steady state formulation. It solves the Reynolds-averaged Navier Stokes set of equations for a
250 turbulent flow ($k - \varepsilon$ model), with a terrain-following structured mesh, allowing also the simulation of forested terrain (Costa et al., 2006) and wind turbine wakes (Gomes et al., 2014; Gomes and Palma, 2016).

### 5.1.1 Integration domain and boundary conditions

The model topography (domain size, $19$ km $\times$ $18.8$ km, around the central location 608250E, 4396621N) was obtained by bi-linear interpolation of terrain data. The positioning of the domain boundaries and its impact on flow variables were part of
255 the work of Silva (2018).

At the inlet a log-law profile was set. To ensure an equilibrium shear stress, the $k$ profile decreases with the square of height above ground level. At the top of the domain a zero shear stress condition was used. The inlet profile's development is caped at the boundary layer's limit, all quantities being constant above that height. At the lateral boundaries a symmetry condition was applied.
The ground was modelled as a rough surface, a wall function, a log-law defining the velocity at the node closest to the ground, and the turbulence model quantities, $k$ and $\varepsilon$. The values used in the computational model for $z_0$ (roughness length) and $u_*$ (friction velocity) were 0.1 (indicated by Wagner et al. (2019) as the roughness length near the double ridge area after conversion from the CORINE Land Cover classes) and 0.25. These values were uniform for the whole domain. The surface cover (forest patches and height) and its representation in the computational model (roughness length, leaf area index, use of a
canopy model) is still a work in progress, as the presence of eucalyptus and pine tree patches in the area are expected to have an impact on the flow. However, this would increase the number of variables influencing the flow results, masking the effects of the digital terrain model alone. See for instance, the effects of forest resolution and wind orientation relative to the forest stands in the computational modelling of flow over forests in Costa et al. (2006).

## 5.2 Computational meshes

A total of 18 computational meshes (Table 5) was used. The central part of the domain ($4\,\text{km} \times 6\,\text{km}$, based on ALS and Mil terrain data), was resolved with uniform horizontal resolution ($20\,\text{m} \times 20\,\text{m}$, $40\,\text{m} \times 40\,\text{m}$ and $80\,\text{m} \times 80\,\text{m}$), expanding towards the domain boundaries with factors $f_x$ and $f_y$ close to 1, to minimise the discretisation errors. The domain's height (3000 m) was discretised by 100 nodes ($N_k$), with the first node 2 m above ground level and a grid expansion $f_z = 1.0435$, yielding a maximum cell size ($\Delta_z$) equal to 124 m. For availability of meshes ALS.SW.## and ALS.NE.##, see Palma et al. (2020) and information under *Data availability*, at the end of this study.

A preliminary analysis showed that the flow variables had low sensitivity to the number of nodes in the vertical ($n_k$), opposed to the height of the first node above ground level, which showed a significant impact and is worthy of further studies.

Three types of meshes were used: SRTM, with the whole domain based on the SRTM data; Mil, a combination of SRTM and Military maps; and ALS, based on all three DTM sources (Figure 5). A minimum of 8818 iterations and 3.87 hours of computing time were required, and a maximum of 20033 iterations and $50\times$ more computing time in the case of mesh Mil.NE.20. Number of iteration is a better indicator of the actual computing time, since the value given here was influenced by the computer load at the time of the calculations.

## 5.3 Flow pattern

The flow modelling analysis was based on the flow patterns at two transects in the case of SW and NE winds (Figures 12 and 13) and wind speed, wind direction and turbulent kinetic energy results for SW (Figures 14–16).

As expected, the flow pattern (Figures 12 and 13) is characterised by separated flow regions in the leeside of either ridges. The figures are coloured by the spanwise velocity component ($v$), showing two different streams: up-valley on the leeside of SW ridge and down-valley on the upwind side of NE ridge (Figure 12) and down-valley in the case of NE winds (Figure 13). The ridge height increases with the grid resolution (see insets) and the detachment point moves to higher elevations, yielding longer and deeper separated flow regions, see for instance Figure 13.

Menke et al. (2019), in their analysis of the experimental data, reported average length and depth equal to 697 and 157 m for both SW and NE wind directions and stratification levels based on the gradient Richardson number between -1 and 1; in the case of neutral flow, length and height equal to 807 and 192 m were reported for a 10-min period. The length and height of the separation zone, in Table 6, tend to increase with the grid refinement (with the exception of the SW winds when refining from 40 to 20 m resolution), predicting a recirculation region longer and narrower compared with the measurements.

## 5.4 Southwesterly winds

The wind speed, wind direction and turbulent kinetic energy profiles at towers 20/tse04, 25/tse09 and 29/tse13 (Figures 14, 15 and 16) show a good agreement of the wind direction with the measurements, a poor agreement of the wind speed (underprediction) at all towers, and underprediction of the turbulent kinetic energy in the valley and at the NE ridge (towers 25/tse09 and 29/tse13, Figures 15c and 16c). A good agreement between computational and experimental results is not expected, mainly

**Table 5.** Computational meshes ($\Delta z_{Min} = 2\,\text{m}$).

| | Name | $\Delta_{x/y}$ min (m) | Max (m) | $N_i \times N_j$ | $t_{CPU}$ (h) | $N_{iter}$ |
|---|---|---|---|---|---|---|
| 1 | SRTM.SW.80 | 80 | 478.6 | 120×155 | 6.27 | 8818 |
| 2 | SRTM.SW.40 | 40 | 400.0 | 200×270 | 52.96 | 11557 |
| 3 | SRTM.SW.20* | 20 | 414.0 | 320×470 | - | - |
| 4 | Mil.SW.80 | 80 | 478.6 | 120×155 | 18.13 | 8906 |
| 5 | Mil.SW.40 | 40 | 400.0 | 200×270 | 89.64 | 11040 |
| 6 | Mil.SW.20 | 20 | 414.0 | 320×470 | 237.35 | 14095 |
| 7 | ALS.SW.80 | 80 | 478.6 | 120×155 | 3.87 | 8898 |
| 8 | ALS.SW.40 | 40 | 400.0 | 200×270 | 15.09 | 10996 |
| 9 | ALS.SW.20 | 20 | 414.0 | 320×470 | 111.28 | 16606 |
| 10 | SRTM.NE.80 | 80 | 478.6 | 120×155 | 19.92 | 9554 |
| 11 | SRTM.NE.40 | 40 | 400.0 | 200×270 | 110.04 | 14227 |
| 12 | SRTM.NE.20 | 20 | 414.0 | 320×470 | 242.57 | 15188 |
| 13 | Mil.NE.80 | 80 | 478.6 | 120×155 | 28.43 | 9313 |
| 14 | Mil.NE.40 | 40 | 400.0 | 200×270 | 101.90 | 13351 |
| 15 | Mil.NE.20 | 20 | 414.0 | 320×470 | 191.38 | 20033 |
| 16 | ALS.NE.80 | 80 | 478.6 | 120×155 | 3.74 | 9322 |
| 17 | ALS.NE.40 | 40 | 400.0 | 200×270 | 93.49 | 11937 |
| 18 | ALS.NE.20 | 20 | 414.0 | 320×470 | 80.21 | 18634 |

| Expansion factors | | | |
|---|---|---|---|
| $\min(\Delta_{x/y})$ | $f_x$ | $f_y$ | $f_z$ |
| 80 | 1.0524 | 1.0331 | 1.0435 |
| 40 | 1.0471 | 1.0299 | 1.0435 |
| 20 | 1.0518 | 1.0271 | 1.0435 |

* Did not meet residual criteria

because of the uniform roughness length; the important is the sensitivity of the computational results to the different numerical meshes.

As an indicator of stationarity, the mean values of the experimental results over the 30-min period are plotted, showing the minima and maxima within that period as error bars. Departure from stationarity condition reaches a higher magnitude in the valley (Figure 15), given the location of tower 25/tse09 inside the recirculation zone. Unsteadiness is a well-known characteristic of recirculation zones and their prediction is very sensitive to spatial resolution (Castro et al., 2003) and terrain model as shown by Figure 15b. The separated flow region, tower 25/tse09 (Figure 15) is characterised by low wind speed

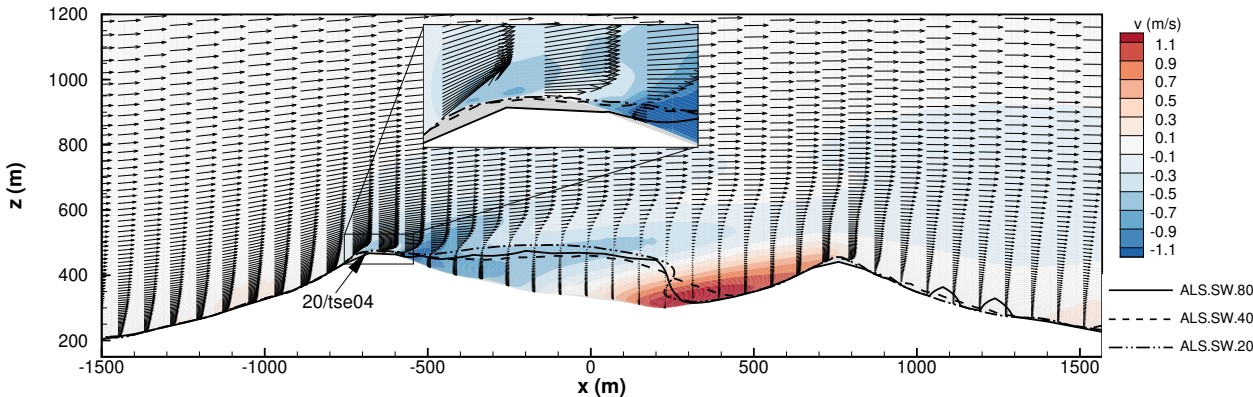

**Figure 12.** Impact of mesh resolution on separation zone in transect that crosses tower 20/tse04, in the case of SW winds.

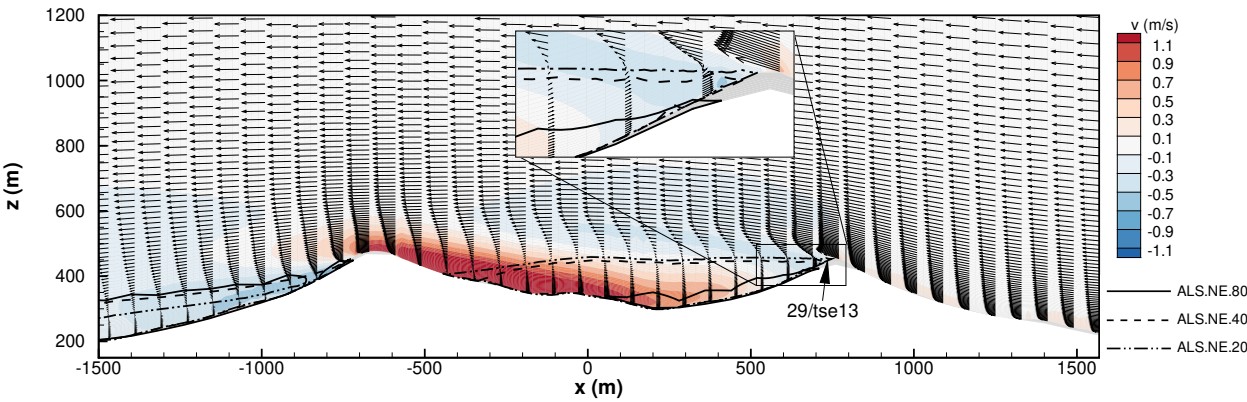

**Figure 13.** Impact of mesh resolution on separation zone in transect that crosses tower 29/tse13, in the case of NE winds.

and rotation of the wind with the distance above the ground. The wind speeds at $z_{asl} > 100$ decreases as the mesh is refined and the height of the recirculation zone increases (Table 6). The flow in the valley is aligned with the valley and therefore

perpendicular to the ridges and the incoming wind. The predicted wind direction is in close agreement with the measurements, with the exception of 40 and 20 m meshes based on Mil DTM.

   As a whole, the flow results appear to be more sensitive to the resolution than to the DTM; see, for instance, Figure 14a) with the results on 40 and 20 m meshes detached from results on the 80 m mesh. At least a resolution of 40 m is required.

   The profiles (not shown) in the case of NE winds (45°) are similar to SW winds, apart from the situation being reversed,

since in this case the first and second ridge are the NE and SW ridges.

**Table 6.** Length and maximum depth of separation zone.

| | Length (m) | Height (m) |
|---|---|---|
| | Southwesterly winds | |
| 80 | 1040.1 | 142.9 |
| 40 | 1120.0 | 131.8 |
| 20 | 1000.1 | 155.0 |
| | Northeasterly winds | |
| 80 | 762.5 | 135.1 |
| 40 | 1120.9 | 143.4 |
| 20 | 1159.3 | 151.8 |

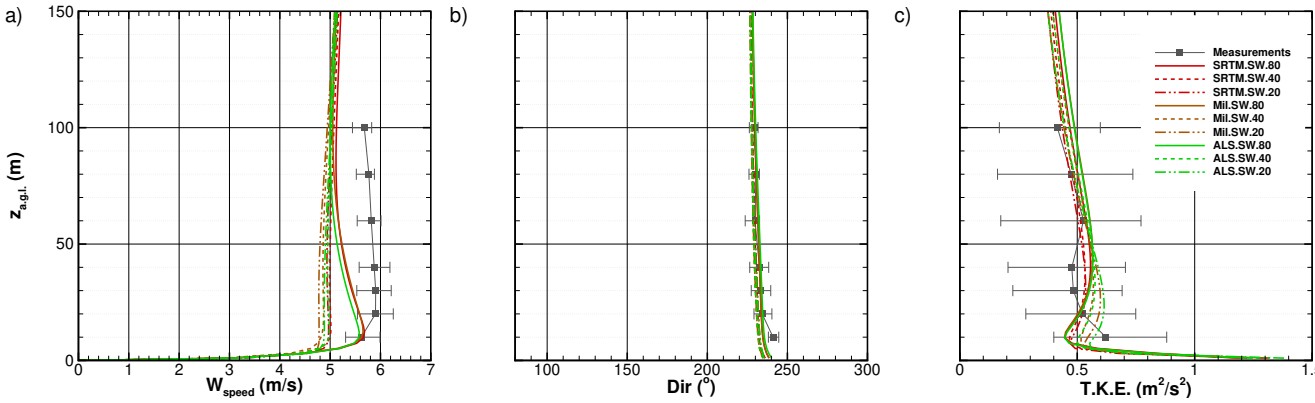

**Figure 14.** Simulation results and experimental data in tower 20/tse04 for SW winds: a) wind speed; b) wind direction; c) turbulent kinetic energy.

Differences between the profiles and the reference profile $ALS_{20}$ were measured in terms of RMSE (Tables 7, 8 and 9), which, in general, show a pattern similar to the slope (Table 4), where the RMSE decreases either by refining the mesh or for a given mesh by moving from the SRTM, to Mil and ALS based meshes. RMSE values at towers 20/tse04 and 29/tse13 (Tables 7 and 9) on the hills depend on the dominant wind directions, showing the effects of the valley flow on the downstream hill. The effect of calculations on 20 m mesh compared to those on 40 m mesh are less than the effect of calculations on 40 m mesh compared to those on 80 m mesh.

## 6   Discussion and conclusions

Meshes for computational modelling of flow over the Perdigão site were created, based on three digital terrain models: high-resolution (2 m resolution) airborne lidar survey (ALS), Military (10 m) and SRTM (30 m) data. The mesh appraisal was

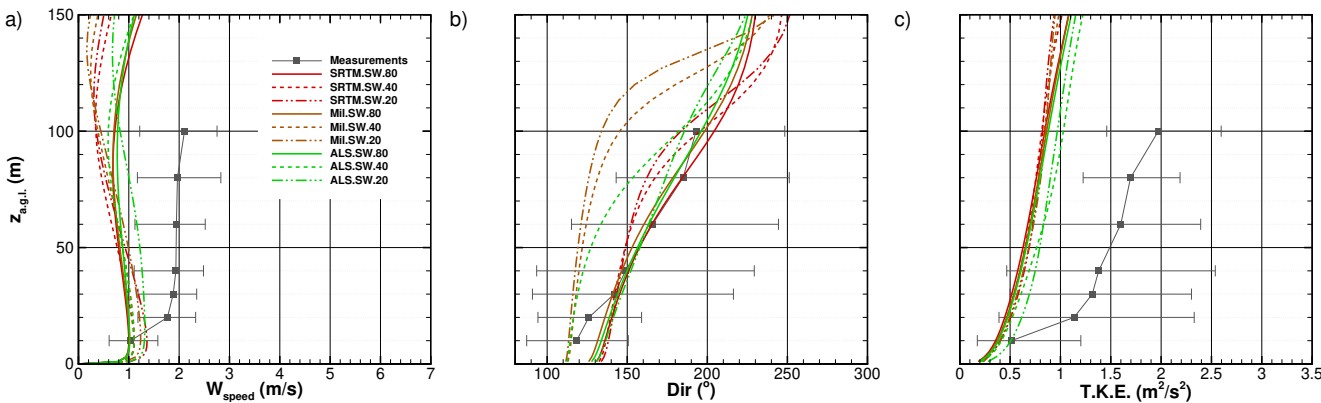

**Figure 15.** Simulation results and experimental data in tower 25/tse09 for SW winds: a) wind speed; b) wind direction; c) turbulent kinetic energy.

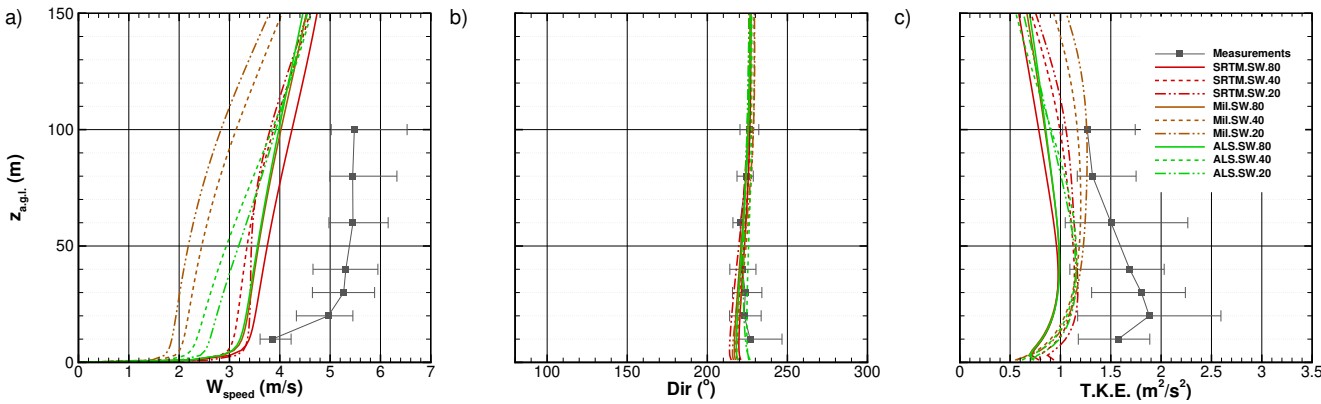

**Figure 16.** Simulation results and experimental data in tower 29/tse13 for SW winds: a) wind speed; b) wind direction; c) turbulent kinetic energy.

carried out in two ways: by their ability to replicate the two main terrain attributes, elevation and slope, and by their effect on the wind flow computational results (wind speed, wind direction and turbulence kinetic energy) at three locations.

About the digital terrain models, the main conclusions were the following:

1. The SRTM data is not an accurate representation of the Perdigão site.

2. Only meshes based on the ALS have the ability to reproduce the smaller scales between 10 and 100 m.

3. The ALS data yielded the lowest elevation errors; average RMSE around 5.8 m on 80 m, decreasing to 0.6 m on 10 m mesh.

**Table 7.** RMSE of wind speed, wind direction and turbulent kinetic energy for tower 20/tse04.

| | $W_{speed}$ (m/s) | | | Dir (°) | | | T.K.E (m²/s²) | | |
|---|---|---|---|---|---|---|---|---|---|
| | SRTM | Mil | ALS | SRTM | Mil | ALS | SRTM | Mil | ALS |
| | Southwesterly winds | | | | | | | | |
| 80 | 0.83 | 0.81 | 0.68 | 4.60 | 4.65 | 6.28 | 0.13 | 0.14 | 0.13 |
| 40 | 0.20 | 0.17 | 0.12 | 0.79 | 1.70 | 1.60 | 0.12 | 0.08 | 0.07 |
| 20 | 0.09 | 0.21 | – | 0.71 | 1.80 | – | 0.11 | 0.03 | – |
| | Northeasterly winds | | | | | | | | |
| 80 | 1.28 | 2.53 | 2.43 | 16.80 | 16.49 | 14.90 | 0.78 | 1.01 | 1.04 |
| 40 | 0.99 | 0.50 | 0.22 | 9.84 | 9.83 | 4.09 | 0.46 | 0.33 | 0.39 |
| 20 | 0.18 | 0.30 | – | 5.04 | 2.74 | – | 0.06 | 0.08 | – |

**Table 8.** RMSE of wind speed, wind direction and turbulent kinetic energy for tower 25/tse09.

| | $W_{speed}$ (m/s) | | | Dir (°) | | | T.K.E (m²/s²) | | |
|---|---|---|---|---|---|---|---|---|---|
| | SRTM | Mil | ALS | SRTM | Mil | ALS | SRTM | Mil | ALS |
| | Southwesterly winds | | | | | | | | |
| 80 | 0.87 | 0.77 | 0.69 | 26.67 | 20.10 | 14.81 | 0.33 | 0.29 | 0.26 |
| 40 | 0.92 | 0.90 | 0.70 | 38.14 | 75.35 | 48.96 | 0.36 | 0.34 | 0.16 |
| 20 | 0.81 | 0.84 | - | 37.03 | 92.73 | - | 0.36 | 0.32 | - |
| | Northeasterly winds | | | | | | | | |
| 80 | 2.82 | 3.67 | 3.54 | 157.07 | 158.04 | 153.56 | 0.25 | 0.49 | 0.49 |
| 40 | 0.39 | 0.88 | 0.53 | 88.82 | 79.37 | 69.95 | 0.51 | 0.33 | 0.23 |
| 20 | 0.19 | 0.17 | – | 40.61 | 43.20 | – | 0.22 | 0.15 | – |

4. The RMSE for SRTM does not go below 7.4 m. A 40 m horizontal resolution based on the ALS data is enough to achieve an error below 1.4 m in five key locations and below 0.28 m using a 20 m mesh.

5. The maximum terrain slope was about $1.8\times$ higher ($-67.81°$) on a 20 m mesh resolution compared with an 80 m mesh resolution ($-37.33°$). An 80 m mesh does not accurately represent elevation and slope, mainly near the extreme elevation values (highs and lows).

The effect of the terrain model on the wind speed, wind direction and turbulent kinetic energy at three locations (SW ridge, valley and NW ridge) and two incoming wind directions (SW and NE) were the following:

1. In the case of SW winds, the mesh resolution effects on the SW ridge were restricted to the first 100 m a.g.l., where mesh refinement decreased the wind speed and degraded the quantitative agreement with the experimental data, though replicating the profile shape.

**Table 9.** RMSE of wind speed, wind direction and turbulent kinetic energy for tower 29/tse13.

| | $W_{speed}$ (m/s) | | | Dir (°) | | | T.K.E (m$^2$/s$^2$) | | |
|---|---|---|---|---|---|---|---|---|---|
| | SRTM | Mil | ALS | SRTM | Mil | ALS | SRTM | Mil | ALS |
| | | | Southwesterly winds | | | | | | |
| 80 | 1.19 | 0.82 | 0.77 | 4.87 | 7.16 | 5.92 | 0.33 | 0.28 | 0.27 |
| 40 | 0.51 | 1.68 | 0.44 | 6.30 | 8.45 | 6.05 | 0.16 | 0.52 | 0.10 |
| 20 | 0.83 | 2.33 | - | 10.40 | 10.03 | - | 0.28 | 0.70 | - |
| | | | Northeasterly winds | | | | | | |
| 80 | 0.44 | 0.49 | 0.44 | 16.15 | 17.73 | 16.79 | 0.13 | 0.15 | 0.13 |
| 40 | 0.26 | 0.16 | 0.24 | 6.94 | 10.26 | 5.35 | 0.06 | 0.07 | 0.06 |
| 20 | 0.18 | 0.09 | – | 2.98 | 4.66 | – | 0.04 | 0.03 | – |

2. Separated flow field in the valley is perpendicular to the main flow direction. This region increases in height and length with the mesh refinement.

3. The flow (mainly the wind direction) in the valley was the most affected by terrain resolution; low velocities (about $1\,\mathrm{m\,s^{-1}}$) are associated with large variations of wind direction within the first 150 m a.g.l..

Concerning the digital terrain models and meshes, the conclusions were the following.

1. It was found that 40 and 20 m meshes are resolutions –threshold resolution– beyond which no or insignificant changes occur both in terrain attributes, elevation and slope, and in the flow field variables, wind speed, wind direction and turbulent kinetic energy.

2. It is recommended that at least 40 and 20 m meshes based on military and ALS be used to describe the Perdigão site, with SRTM restricted to far away regions.

The conclusions hold under the conditions of the present work, namely terrain data and flow model equations and conditions. Under different conditions, further validation may be required.

*Data availability.* Three datafile types are available. For more information see Palma et al. (2020)

**Aerial survey files (as described in section 2.1)** :

1. Ortophotos in $5\,\mathrm{cm}$ and $20\,\mathrm{cm}$ resolution
https://perdigao.fe.up.pt/datasets/thredds/catalog/landCharacterization/Aerial%20Survey%20Lidar%20and%20Photography%20Data/Images/catalog

2. Lidar point cloud data

 https://perdigao.fe.up.pt/datasets/thredds/catalog/landCharacterization/Aerial%20Survey%20Lidar%20and%20Photography%20Data/Pointcloud/catalog

**Digital Terrain Models in local metric datum (as described in section 4)** :

1. SRTM raster map of Perdigão region, $\sim 100\,\mathrm{km}$ square area at resolution of 1 arc-second ($\approx 24\,\mathrm{m} \times 31\,\mathrm{m}$, Easting $\times$ Northing) (non-uniform)

    https://windsptds.fe.up.pt//thredds/fileServer/landCharacterization/Digital%20Terrain%20Models/dtm_srtm_1arcsec.zip

2. Military charts raster map of Perdigão region, $16\,\mathrm{km} \times 20\,\mathrm{km}$ area at $10\,\mathrm{m}$ resolution

    https://windsptds.fe.up.pt//thredds/fileServer/landCharacterization/Digital%20Terrain%20Models/dtm_mil_10m.zip

3. ALS derived raster maps of Perdigão site, $\sim 5\,\mathrm{km} \times 6\,\mathrm{km}$ area (net) at $2\,\mathrm{m}$ resolution, without buildings

    https://windsptds.fe.up.pt//thredds/fileServer/landCharacterization/Digital%20Terrain%20Models/dtm_als_no_buildings_2m.zip:

4. ALS derived raster maps of Perdigão site, $\sim 5\,\mathrm{km} \times 6\,\mathrm{km}$ area (net) at $2\,\mathrm{m}$ resolution with buildings

    https://windsptds.fe.up.pt//thredds/fileServer/landCharacterization/Digital%20Terrain%20Models/dtm_als_buildings_2m.zip

**Computational meshes (as described in section 5.2)** :

1. NE inflow, $20\,\mathrm{m} \times 20\,\mathrm{m}$ (ALS.NE.20, as in Table 5)

    https://windsptds.fe.up.pt//thredds/fileServer/landCharacterization/Computational%20Topography%20Meshes/mesh_ne_20x20.dat

2. NE inflow, $40\,\mathrm{m} \times 40\,\mathrm{m}$ (ALS.NE.40, as in Table 5)

    https://windsptds.fe.up.pt//thredds/fileServer/landCharacterization/Computational%20Topography%20Meshes/mesh_ne_40x40.dat

3. NE inflow, $80\,\mathrm{m} \times 80\,\mathrm{m}$ (ALS.NE.80, as in Table 5)

    https://windsptds.fe.up.pt//thredds/fileServer/landCharacterization/Computational%20Topography%20Meshes/mesh_ne_80x80.dat

4. SW inflow, $20\,\mathrm{m} \times 20\,\mathrm{m}$ (ALS.SW.20, as in Table 5)

    https://windsptds.fe.up.pt//thredds/fileServer/landCharacterization/Computational%20Topography%20Meshes/mesh_sw_20x20.dat

5. SW inflow, $40\,\mathrm{m} \times 40\,\mathrm{m}$ (ALS.SW.40, as in Table 5)

    https://windsptds.fe.up.pt//thredds/fileServer/landCharacterization/Computational%20Topography%20Meshes/mesh_sw_40x40.dat

6. SW inflow, $80\,\text{m} \times 80\,\text{m}$ (ALS.SW.80, as in Table 5)

   https://windsptds.fe.up.pt//thredds/fileServer/landCharacterization/Computational%20Topography%20Meshes/mesh_ sw_80x80.dat

*Author contributions.* José M.L.M. Palma conceived, coordinated and was responsible for both the work and the manuscript writing. Carlos A.M. Silva carried out the fluid flow calculations, under the guidance of Vitor M.C. Gomes and Alexandre Silva Lopes. Teresa Simões and Paula Costa developed the algorithm for building identification. Vasco T.P. Batista contributed by processing the terrain data using the LAStools software.

*Competing interests.* No competing interests are present.

*Acknowledgements.* The Perdigão-2017 field campaign was primarily funded by the US National Science Foundation, European Commission (ENER/FP7/618122/NEWA), Danish Energy Agency, German Federal Ministry of Economy and Energy, FCT-Portuguese Foundation for Science and Technology (NEWA/1/2014), and US Army Research Laboratory. We are grateful to the municipality of Vila Velha de Ródão, landowners who authorized installation of scientific equipment in their properties, the residents of Vale do Cobrão, Foz do Cobrão, Alvaiade, Chão das Servas and local businesses who kindly contributed to the success of the campaign. The campaign would not have been possible without the alliance of many persons and entities, too many to be listed here and to whom we are also grateful.

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
