# Peer review of "The digital terrain model in the computational modelling of the flow over the Perdigão site: the appropriate grid size"

_Wind Energy Science, 2019_

## Referee Comment (RC1) · Paul van der Laan (Referee) · 30 Mar 2020

**Review of *The digital terrain model in the computational modelling of the flow over the Perdigão site: the appropriate grid size* by José M.L.M. Palma et al.**

Reviewer: M. Paul van der Laan, DTU Wind Energy

The authors investigate the impact of different terrain models with different resolutions on atmospheric flow simulations of the Perdigão site. The authors recommend a horizontal *threshold resolution* of 40 m. The authors define a threshold resolution, as the resolution beyond which the model quality deteriorated quickly, but below which no significant improvement in modelling results was observed. The article is well written, follows a good structure and includes a lot of useful technical information about the Perdigão site and related terrain models. However, I have two main comments about the article.

My main concern about using a *threshold resolution* is the fact that is not general, while it is the main objective of the article to find such a resolution, as the authors mention on Page 3, Line 68. The required resolution of the terrain map and flow model mesh are dependent on the quantities of interest. For example, one could be interested in the integrated drag force of the entire terrain, which might require a less strict level of refinement compared to a specific profile of a flow quantity near the ground and in the lee of a steep hill. This means that the concept of a threshold resolution, as mentioned in the introduction is not general, but it depends on the quantity of interest. In addition, the chosen flow model can also influence the required resolution. This is the reason why every article including a new flow model setup needs to be verified by a grid refinement study.

My second main comment is that not all conclusions are motivated by the presented results because the grid errors of vertical profiles of wind speed, wind direction and turbulent kinetic energy are not directly shown and not clearly quantified.

I have listed related and additional main and minor comments below, which need to addressed in order to consider a publication in Wind Energy Science.

**Main comments**

1. Page 3, Line 54: What is meant by *terrain attributes* and *topographic meaning* in *Deng et al. (2007) indicates that the mesh resolution can change not only terrain attributes in specific points but also the topographic meaning of attributes at each point.* Do you mean roughness length? Please clarify.

2. What do you mean by *switched on 1 sec registration after assistance by the Portuguese National Mapping Agency*? Do you mean a sampling frequency of 1 Hz?

3. Section 5.1.1: I do not understand that you $k$ profile is varying with height. If you use a log profile for the streamwise velocity at the inlet, then I assume that you are modelling an atmospheric surface layer, which is represents a constant $k$ value, as discussed by Richards and Hoxey (1993). In addition, if you model a boundary layer height by simply capping it above a certain height, how do you make sure that such an inflow profile is in balance with an empty (flat terrain) flow domain? If the inflow profile is not in balance with your RANS model, then the results at area of interest are dependent on the distance of inflow boundary to the area of interest, which is highly unfavorable.

4. Page 6, Line 157: You mention *The slope, ... varies between 21.08° and 45.09°, always above the threshold for flow separation (Wood, 1995)*. However, flow separation also depends on the roughness length and atmospheric conditions as turbulence intensity and atmospheric stability. So an attached flow could exist for a hill with a 21°slope if the conditions allow it. Wood (1995) only looked at neutral atmospheric conditions. Therefore, I think you rephrase your statement that flow separation is likely to occur for the site your are investigating.

5. Page 16-17, Lines 272-274: You mention that results from 20 and 40 m meshes yield similar results and appear to be accurate enough for computational modelling of atmospheric flow over Perdigão based on Figure 12, using the reversed flow regions. Please clarify *appear to be accurate enough* by quantifying the differences in order to motivate your statement. You could also remove this statement and quantify the differences in Section 5.4

6. Figures 14-16: I would not plot the simulation results of the different meshes together with the measurements in the same figure in order to separate the grid refinement study (model verification) from the model validation. I would also remove the statement on Page 19, Lines 286-287: *For some reason, in the valley the best agreement with the experimental data occurred in the case of the coarser meshes.* (A common mistake in literature is to choose a coarser grid because it compares better with measurements and I would recommended that you do not suggest the reader to do so.) If you would like to include a model validation, you could make separate plots of the chosen grid size (or finest grid size) for each terrain input model and measurements. In addition, have you tried to normalize the measurements and CFD results of wind speed and TKE with their local friction velocity $u_{*0}$?

7. Page 19, Lines 288-290: You mention: *As a whole, results depend more on the resolution than on the DTM and at least a resolution of 40 m is required. Differences between the computational results on 20 and 40 m resolution meshes are minor and within the uncertainty of computational modelling.* This statement is not sufficiently shown in Figure 14-16. I would suggest to (also) plot the differences between the inflow profiles in percentages as function of height, with respect to the reference simulations. This should provide a more clear presentation of the differences between the simulations compared to the RMSE values of Tables 6-8. You can then conclude that the grid error in (for example) wind speed at the reference locations, at a certain height (e.g. around a typical onshore wind turbine hub height) is x% and then it is easier to quantify the impact of grid resolution and terrain model on a wind resource assessment.

8. Page 20, Line 304: I could not find the second statement of the conclusion elsewhere in the article: *Only meshes based on the ALS have the ability to reproduce the smaller scales between 10 and 100 m.* Please remove the statement from the conclusion or motivate it in the article.

9. Page 21, Lines 320-327: The conclusions made here are not motivated by the results presented in the article.

    (a) I do not agree with the concept of a threshold resolution because it depends on both the applied flow solver and quantity of interest. You can either remove it or reduce the statement by writing that required resolution only applies to your investigated quantities of interest and the chosen flow solver. This also applies to the abstract, title and motivation.

    (b) Statements 2 and 3 should be motivated with plots showing the grid error in terms of percentages, as mention previously.

**Minor comments**

1. Section 1.1: It is more common to use a past tense instead of a present tense when referring to literature. (For example on Page 3, Line 59: develops $\rightarrow$ developed.)

2. Page 3, Line 64: Please rephrase *..the flow with the sharp edge..*, because it is the terrain model geometry that has a sharp edge, not the flow.

3. Page 15, Line 253: I would abbreviate Reynods-averaged Navier-Stokes as RANS, which is more common. In addition, you forgot to define RANS.

4. Page 15, Line 253: I would write the two equation $k$-$\varepsilon$ model with $k$ instead of $\kappa$, as $\kappa$ is commonly used as the Von Karman constant and you also use $k$ as the turbulent kinetic energy.

5. Not all reported values need to reported fully. For example, you could report 22071075 m$^2$ as $22.1 \times 10^6$ m$^2$ (Page 4, Line 79) and 993198375 as $10^9$ approximately (Page 4, Line 79), this also applies elsewhere in the article.

6. Page 6, Line 33: I would rephrase *the parallelism between the two ridges*.

7. Page 7, Line 145: Do you know the distances between the ridges in cm? If not, I would remove the two zeros.

**References**

Richards, P. J. and Hoxey, R. P.: Appropriate boundary conditions for computational wind engineering models using the $k$-$\varepsilon$ turbulence model, Journal of Wind Engineering and Industrial Aerodynamics, 46,47, 145–153, 1993.

---

## Referee Comment (RC2) · Anonymous Referee #2 · 30 Mar 2020

The paper addresses the important problem of the numerical model setup for simulations of the wind over two parallel ridges at the Perdigao site, the area of interest during the intensive observational campaign of the New European Wind Atlas project.

The introduction is too long, given that the problem is quite a straightforward one: the numerical model resolution is sufficient when further refinement no longer affects the results. This is often ignored by necessity due to computational cost, and compromises are generally accepted if the results are still applicable after validation. So the goal of this interesting and relevant paper could be elegantly achieved by presenting a clear and comprehensive overview of the resolution requirements, so that future simulations

of this highly studied area can easily be set up and evaluated with respect to the terrain and grid resolution.

I would recommend that the paper is published after revision, addressing the specific comments below.

Specific comments:

L23: It is unclear what is meant with the resolution of the measuring equipment. Perhaps density would be a better word.

L24: Please provide the geographic coordinates in WGS84 at this stage, and the equivalent in UTM. It is acceptable to use UTM later in the text.

L33: The SRTM terrain resolution is 1" (24 m) but in the conclusions it is considered inadequate, in spite that the 40 m resolution is recommended as a minimum. Please comment.

L58: The threshold resolution is not a concept, but a requirement in numerical modelling. It is often ignored by necessity due to computation cost.

L80: 22 square km would be easier to understand than 22 million square meters.

L79,84: What is BlomTopEye? TPDS? Please supply a reference or description.

L81: This is an impressive number of points in the cloud, but does it have any relevance? Please remove.

L83-89: This paragraph is very hard to comprehend. Please reformulate.

L94, Figure 1: please display the two images in the same projection

L96: Table ?? ?

L135: It is important to remember that the concept of two-dimensionality quickly breaks in the case of atmospheric flows, i.e. as soon as not all the forces acting on an air parcel are aligned with the 2D plane.

L146: It is inappropriate to use anything more precise than integer meters when describing terrain altitude with regard to atmospheric modelling. The same for geographic coordinates, for example in Table 1, especially since the highest data resolution is 2 m.

L228-233, Section 5: It is unlikely that the atmospheric conditions at 22 UTC are neutral and stationary, as the simulation setup assumes. It is also unclear whether the authors are aware that the conditions might not be neutral and stationary, yet they proceed with the analysis because the referred publications have established the opposite? Please clarify by including the measurements of relevant parameters.

L246-250: Is it justifiable to use uniform roughness for this simulation? Have you performed any sensitivity analyses if the sufficient resolution threshold changes when non-uniform roughness is used?

L250: Does the impact of the first node height show any convergence, like it is the case for the horizontal resolution? You state that this is worthy of further studies, whereas this is the very study where this should be evaluated.

L278, Figures 12,13: It is not clear what in these figures exactly illustrates the impact of mesh resolution. Are we supposed to see the differences in the insets? Please provide more information in the captions and some discussion about the figures in the text.

L282: The statement how too high $z_0$ and $u^*$ "yield a high loss of momentum" is probably badly formulated.

L291, Figures 14, 15, 16: There are several revealing features in the figures, but only the absolute wind speed, and to some extent the TKE, receive any discussion. The diverging wind speed in the higher altitudes in Figure 15a is particularly interesting. The legend is hard to read.

L294: Please clarify what is meant by "... where a pattern similar to the slope (Table 4) can be observed."

L294: Is the RMSE calculated over all the points in the vertical where the measurements are available? The relevance of this statistical measure in a situation where the basic characteristics of the two datasets disagree quite fundamentally (e.g. the TKE at the tower 20/tse04, Figure 14) is probably questionable.

L304: This conclusion 1. is very categorical, while not entirely based on presented facts. At the resolution of 40 m, which is suggested as sufficient for Perdigao, the RMSE for the SRTM-based simulation is not systematically larger than of the other two datasets; in fact, it can be seen that in terms of RMSE for the 40 m resolution the SRTM-based simulations often have the lowest RMSE (Tables 6, 7, 8).
* * *

---

## Author Comment (AC1) · 11 Apr 2020

**The digital terrain model in the computational modelling of the flow over the Perdigão site: the appropriate grid size**

authored by

José M. L. M. Palma, Carlos A. M. Silva, Vitor M. C. Gomes, Alexandre Silva Lopes, Teresa Simões, Paula Costa, and Vasco T. P. Batista
Submitted for publication in the *Wind Energy Science*
https://doi.org/10.5194/wes-2019-96

**Reply to comments by Reviewer 1 (M. Paul van der Laan, DTU Wind Energy)**

The authors investigate the impact of different terrain models with different resolutions on atmospheric flow simulations of the Perdigão site. The authors recommend a horizontal threshold resolution of 40 m. The authors define a *threshold resolution*, as the resolution beyond which the model quality deteriorated quickly, but below which no significant improvement in modelling results was observed. The article is well written, follows a good structure and includes a lot of useful technical information about the Perdigão site and related terrain models. However, I have two main comments about the article.

My main concern about using a *threshold resolution* is the fact that is not general, while it is the main objective of the article to find such a resolution, as the authors mention on Page 3, Line 68. The required resolution of the terrain map and flow model mesh are dependent on the quantities of interest. For example, one could be interested in the integrated drag force of the entire terrain, which might require a less strict level of refinement compared to a specific profile of a flow quantity near the ground and in the lee of a steep hill. This means that the concept of a threshold resolution, as mentioned in the introduction is not general, but it depends on the quantity of interest. In addition, the chosen flow model can also influence the required resolution. This is the reason why every article including a new flow model setup needs to be verified by a grid refinement study.

REPLY (First/Main concern or comment):

1. As shown by the title, our main concern is the terrain modelling, not the flow modelling or the flow results. First, we wanted to know to what extent the resolution of the digital terrain model (DTM) affects the terrain modelling and, second, how the terrain modelling affects the flow results.

2. The first (assessment of resolution on terrain modelling, section 4) was carried out by comparing terrain modelling results (terrain elevation and slope) on meshes of different resolutions (80, 40 and 20 m) based on two DTMs of different resolution (SRTM and Mil) against terrain modelling results on meshes based on a higher resolution (ALS) DTM.

   - This analysis was made on the basis of terrain properties (or attributes) of interest to atmospheric flow modelling; i.e. terrain elevation and terrain slope. Because these two are the most important and for the sake of simplicity, land coverage (roughness) was assumed constant.

3. The second (impact of terrain model resolution on flow model results, section 5) was carried out by analysing the flow results based on those meshes and DTMs.

   - This analysis was based on the vertical profiles of wind speed, wind direction and turbulent kinetic energy at three locations, and flow pattern at two transects. These variables and the locations were selected because of their importance in wind energy and sensitivity to the mesh resolution.

4. About *threshold resolution* and flow model results we note the following:

   (a) One must distinguish between the *threshold resolution* in the context of the terrain model and in the context of the flow model. The former is much simpler, independent of the flow model and was dealt with in full in section 4.

   (b) In the case of the flow model results, the *threshold resolution* depends on the flow model (e.g., discretization techniques) and for the same flow model will depend of the parameters of a given run (domain size, boundary conditions, mesh resolution along any direction, etc.).

   (c) The mesh is comprised by the horizontal mesh describing the terrain shape (the digital terrain model), often a terrain-following coordinate system, and the mesh along the vertical.

   (d) There are guidelines for setting the vertical mesh, namely the distance between the first grid node and the ground, but there are no guidelines for how fine the horizontal mesh should be.

   (e) The comment that *threshold resolution* is not general and depends on the flow model, suggests that our conclusions cannot be generalised. We are aware of that, nevertheless, we consider ours a useful work, because we are not expecting the resolution requirements (or the *threshold resolution*) to be much different, given the similarities among the available models.

My second main comment is that not all conclusions are motivated by the presented results because the grid errors of vertical profiles of wind speed, wind direction and turbulent kinetic energy are not directly shown and not clearly quantified.

REPLY (Second main comment):
We do not understand this comment. The vertical profiles of wind speed, wind direction and turbulent kinetic energy are shown (Figures 14, 15 and 16) for all meshes (80, 40 and 20 m) and DTMs (SRTM, Mil and ALS) and the errors quantified (see RMSE in Tables 6, 7 and 8).

I have listed related and additional main and minor comments below, which need to [be] addressed in order to consider a publication in Wind Energy Science.

**Main comments**

1. Page 3, Line 54: What is meant *by terrain attributes and topographic meaning in Deng et al. (2007) indicates that the mesh resolution can change not only terrain attributes in*

*specific points but also the topographic meaning of attributes at each point.* Do you mean roughness length? Please clarify.

> REPLY:
> Terrain attributes are elevation, slope, plan and profile curvature, and topographic wetness index, as defined in lines 45 and 46. For instance, Deng et al. (2007) listed and evaluated six terrain attributes (slope, plan curvature, profile curvature, north-south slope orientation, east-west slope orientation, and topographic wetness index) as a function of DTM resolution.
> The terrain attributes of interest to atmospheric flow modelling are terrain elevation, terrain slope and terrain or land coverage, i.e. roughness.

2. What do you mean by *switched on 1 sec registration after assistance by the Portuguese National Mapping Agency*? Do you mean a sampling frequency of 1 Hz?

> REPLY:
> Yes.

3. Section 5.1.1: I do not understand that you $k$ profile is varying with height. If you use a log profile for the streamwise velocity at the inlet, then I assume that you are modelling an atmospheric surface layer, which is represents a constant $k$ value, as discussed by Richards and Hoxey (1993). In addition, if you model a boundary layer height by simply capping it above a certain height, how do you make sure that such an inflow profile is in balance with an empty (flat terrain) flow domain? If the inflow profile is not in balance with your RANS model, then the results at area of interest are dependent on the distance of inflow boundary to the area of interest, which is highly unfavorable.

> REPLY:
> The distance between the inlet boundary and the area of interest was identical in all simulations.

4. Page 6, Line 157: You mention *The slope, ... varies between 21.08° and 45.09°, always above the threshold for flow separation (Wood, 1995).* However, flow separation also depends on the roughness length and atmospheric conditions as turbulence intensity and atmospheric stability. So an attached flow could exist for a hill with a 21° slope if the conditions allow it. Wood (1995) only looked at neutral atmospheric conditions. Therefore, I think you rephrase your statement that flow separation is likely to occur for the site your are investigating.

> REPLY:
> Changed, as suggested. The new version reads:
> *The slope $(S = |atan(h_{SW,NE}/2)|/\ell_{SW,NE})$, also on a $20\,$m grid varies between $21.08°$ and $45.09°$, always above the threshold for flow separation under neutral conditions (Wood, 1995).*

5. Page 16-17, Lines 272-274: You mention that results from 20 and 40 m meshes yield similar results and appear to be accurate enough for computational modelling of atmospheric flow

over Perdigão based on Figure 12, using the reversed flow regions. Please clarify *appear to be accurate enough* by quantifying the differences in order to motivate your statement. You could also remove this statement and quantify the differences in Section 5.4.

REPLY:
The text was rewritten and experimental values by Menke et al. (2019) included.

6. Figures 14-16: I would not plot the simulation results of the different meshes together with the measurements in the same figure in order to separate the grid refinement study (model verification) from the model validation. I would also remove the statement on Page 19, Lines 286-287: *For some reason, in the valley the best agreement with the experimental data occurred in the case of the coarser meshes.* (A common mistake in literature is to choose a coarser grid because it compares better with measurements and I would recommended that you do not suggest the reader to do so.) If you would like to include a model validation, you could make separate plots of the chosen grid size (or finest grid size) for each terrain input model and measurements. In addition, have you tried to normalize the measurements and CFD results of wind speed and TKE with their local friction velocity $u_{*0}$?

REPLY:
Computational and experimental results in the same figure are necessary to show how similar or different they are. Plots of the chosen grid size for each terrain input would increase the number of figures. By plotting the results at the valley we are not suggesting that the reader should choose the coarser meshes; cases like these, where coarser meshes provide better agreement with experimental data, are not so uncommon. The text was rewritten.

7. Page 19, Lines 288-290: You mention: *As a whole, results depend more on the resolution than on the DTM and at least a resolution of 40 m is required. Differences between the computational results on 20 and 40 m resolution meshes are minor and within the uncertainty of computational modelling.* This statement is not sufficiently shown in Figure 14-16. I would suggest to (also) plot the differences between the inflow profiles in percentages as function of height, with respect to the reference simulations. This should provide a more clear presentation of the differences between the simulations compared to the RMSE values of Tables 6-8. You can then conclude that the grid error in (for example) wind speed at the reference locations, at a certain height (e.g. around a typical onshore wind turbine hub height) is x% and then it is easier to quantify the impact of grid resolution and terrain model on a wind resource assessment.

REPLY:
First, we do not see how the suggested representation can provide a clearer presentation of the differences between the simulations. Second, this type of representation (in absolute values) gives us an idea of the uncertainty for each variable in its own absolute values.

8. Page 20, Line 304: I could not find the second statement of the conclusion elsewhere in the article: *Only meshes based on the ALS have the ability to reproduce the smaller scales between 10 and 100 m.* Please remove the statement from the conclusion or motivate it in the article.

REPLY:
This is related to Figure 11 and section 4.4.

9. Page 21, Lines 320-327: The conclusions made here are not motivated by the results presented in the article.

   (a) I do not agree with the concept of a threshold resolution because it depends on both the applied flow solver and quantity of interest. You can either remove it or reduce the statement by writing that required resolution only applies to your investigated quantities of interest and the chosen flow solver. This also applies to the abstract, title and motivation.

      REPLY:
      Please, refer to our first comment on terrain attributes and objective of the study. Note also that the title is already a long one and the conditions under which our study was carried out are described in full in the abstract, including the flow solver.

   (b) Statements 2 and 3 should be motivated with plots showing the grid error in terms of percentages, as mention previously.

      REPLY:
      This matter is the subject of section 4, thoroughly illustrated by Figures 6 to 11, and Tables 3 and 4. The statement (2) that SRTM should be restricted to far away regions is supported by Table 3, where it can be seen that the absolute error of SRTM is equal to 10 m. Statement 3 is a final recommendation and message of the article, where we say, please use this dataset (ALS) and meshes of at least 40 m horizontal resolution.

**Minor comments**

1. Section 1.1: It is more common to use a past tense instead of a present tense when referring to literature. (For example on Page 3, Line 59: develops → developed.)

   REPLY:
   You are right. The whole section 1.1 was revised.

2. Page 3, Line 64: Please rephrase *..the flow with the sharp edge..*, because it is the terrain model geometry that has a sharp edge, not the flow.

   REPLY:
   It now reads *the cliff with the sharp edge*

3. Page 15, Line 253: I would abbreviate Reynolds-averaged Navier-Stokes as RANS, which is more common. In addition, you forgot to define RANS.

   REPLY:
   The acronym was removed, because it was not used.

4. Page 15, Line 253: I would write the two equation $k - \varepsilon$ model with $k$ instead of $\kappa$, as $\kappa$ is commonly used as the Von Karman constant and you also use $k$ as the turbulent kinetic energy.

   REPLY:
   This was a misspelling. It should be as in line 247.

5. Not all reported values need to reported fully. For example, you could report 22071075 m$^2$ as $22.1 \times 10^6$ m$^2$ (Page 4, Line 79) and 993198375 as $10^9$ approximately (Page 4, Line 79), this also applies elsewhere in the article.

   REPLY:
   These are detailed technical specifications, usually found in aerial topographical surveys, useful to readers in this area.

6. Page 6, Line 33: I would rephrase the parallelism between the two ridges.

   REPLY:
   We do not see anything wrong with this sentence.

7. Page 7, Line 145: Do you know the distances between the ridges in cm? If not, I would remove the two zeros.

   REPLY:
   Yes, you are right.

**References**

Y. Deng, J. P. Wilson, and B. O. Bauer. DEM resolution dependencies of terrain attributes across a landscape. *International Journal of Geographical Information Science*, 21(2):187–213, January 2007. ISSN 1365-8816, 1362-3087. doi: 10.1080/13658810600894364. URL http://www.tandfonline.com/doi/abs/10.1080/13658810600894364.

R. Menke, N. Vasiljević, J. Mann, and J. K. Lundquist. Characterization of flow recirculation zones at the Perdigão site using multi-lidar measurements. *Atmospheric Chemistry and Physics*, 19(4):2713–2723, March 2019. ISSN 1680-7316. doi: 10.5194/acp-19-2713-2019. URL https://www.atmos-chem-phys.net/19/2713/2019/.

P.J Richards and R.P. Hoxey. Appropriate boundary conditions for computational wind engineering models using the $k - \varepsilon$ turbulence model. *Journal of Wind Engineering and Industrial Aerodynamics*, 46-47:145–153, August 1993. ISSN 0167-6105. doi: 10.1016/0167-6105(93) 90124-7. URL http://www.sciencedirect.com/science/article/B6V3M-4847B4C-K/2/6f6fa2909840b477522ec73bd33ba220.

N. Wood. The onset of separation in neutral, turbulent flow over hills. *Boundary-Layer Meteorology*, 76(1-2):137–164, 1995. ISSN 0006-8314. doi: 10.1007/BF00710894. URL http://www.springerlink.com/content/v2412rh223072568/.

---

## Author Comment (AC2) · 11 Apr 2020

**The digital terrain model in the computational modelling of the flow over the Perdigão site: the appropriate grid size**

authored by

José M. L. M. Palma, Carlos A. M. Silva, Vitor M. C. Gomes, Alexandre Silva Lopes, Teresa Simões, Paula Costa, and Vasco T. P. Batista
Submitted for publication in the *Wind Energy Science*
https://doi.org/10.5194/wes-2019-96

**Reply to comments by Reviewer 2 (Anonymous)**

The paper addresses the important problem of the numerical model setup for simulations of the wind over two parallel ridges at the Perdigão site, the area of interest during the intensive observational campaign of the New European Wind Atlas project.

The introduction is too long, given that the problem is quite a straightforward one: the numerical model resolution is sufficient when further refinement no longer affects the results. This is often ignored by necessity due to computational cost, and compromises are generally accepted if the results are still applicable after validation. So the goal of this interesting and relevant paper could be elegantly achieved by presenting a clear and comprehensive overview of the resolution requirements, so that future simulations of this highly studied area can easily be set up and evaluated with respect to the terrain and grid resolution.

> REPLY:
> We do not understand the comment. The introduction is two pages long. This is not a grid refinement study as usually in a computational fluid dynamics study. We are concerned mainly with the resolution of the digital terrain model. These are matters that go beyond computational fluid dynamics and have been dealt with in other disciplines, namely geomorphology. We are of the opinion that mainly in complex terrain, the accuracy of terrain model should precede the CFD three-dimensional grid and this is the message that we try to convey in section 1. Furthermore, the final message is that in Perdigão (and any complex site) the publicly available databases (e.g. SRTM, ASTER) are not accurate enough.

I would recommend that the paper is published after revision, addressing the specific comments below.

Specific comments:

L23: It is unclear what is meant with the resolution of the measuring equipment. Perhaps density would be a better word.

> REPLY:
> The proximity between measuring stations is very low. It reads now *on par with the resolution provided by such a large number of measuring equipment within a small region.*

L24: Please provide the geographic coordinates in WGS84 at this stage, and the equivalent in UTM. It is acceptable to use UTM later in the text.

REPLY:
Yes.

L33: The SRTM terrain resolution is 1" (24 m) but in the conclusions it is considered inadequate, in spite that the 40 m resolution is recommended as a minimum. Please comment.

REPLY:
For meshes with identical horizontal resolution (Figures 6 and 7), both SRTM and Mil yield higher RMSE of terrain elevation than meshes based on ALS. Terrain elevation ($z_{Max}$, Table 3) and slope ($S_{Max}$, Table 4) are higher for meshes based on ALS. It is not just a question of horizontal resolution, but the RMSE of terrain elevation of both SRTM and Mil. The abstract says it all.

L58: The threshold resolution is not a concept, but a requirement in numerical modelling. It is often ignored by necessity due to computation cost.

REPLY:
No comment.

L80: 22 square km would be easier to understand than 22 million square meters.

REPLY:
No comment.

L79,84: What is Blom TopEye? TPDS? Please supply a reference or description.

REPLY:
Blom[1] is the largest company in Europe, specialized on topographic laser ranging and scanning (Mallet and Bretar, 2009). TPDS and TASQ are acronyms of software by Blom. The information in the first four paragraphs of section 2.1 is technical information related to the lidar scanning.

L81: This is an impressive number of points in the cloud, but does it have any relevance? Please remove.

REPLY:
This information is generally provided in all laser scanning surveys and is essential for determining the horizontal resolution.
* * *
[1]http://www.skgeodesy.sk/files/slovensky/ugkk/medzinarodna-spolupraca/bilateralna-spolupraca/norSirotek-2-Bratislava.pdf

L83-89: This paragraph is very hard to comprehend. Please reformulate.

    REPLY:
      The text was rewritten.

L94, Figure 1: please display the two images in the same projection

    REPLY:
    Our apologies, we were unable to display the images in the same projection, due to different software used to draw either image.

L96: Table ?? ?

    REPLY:
    Reference to a missing table was removed.

L135: It is important to remember that the concept of two-dimensionality quickly breaks in the case of atmospheric flows, i.e. as soon as not all the forces acting on an air parcel are aligned with the 2D plane.

    REPLY:
    Yes, but that does not invalidate our statement, which is from the strictly geometrical point of view, given the high length to width ratio.

L146: It is inappropriate to use anything more precise than integer meters when describing terrain altitude with regard to atmospheric modelling. The same for geographic coordinates, for example in Table 1, especially since the highest data resolution is 2 m.

    REPLY:
    The 2 m resolution is the horizontal resolution, because this was the finest mesh generated from the lidar data. However, it is acknowledged that the vertical and planimetric accuracy of the airborne laser scanning are lower than 0.1 m and 0.4 cm (Mallet and Bretar, 2009). Geographic coordinates are now in metre (Table 1) and terrain elevation in decimetre (Tables 1 and 2).

L228-233, Section 5: It is unlikely that the atmospheric conditions at 22 UTC are neutral and stationary, as the simulation setup assumes. It is also unclear whether the authors are aware that the conditions might not be neutral and stationary, yet they proceed with the analysis because the referred publications have established the opposite? Please clarify by including the measurements of relevant parameters.

REPLY:

Yes, we are aware. That was the reason why we selected the 30 minutes averaged between 22:09–22:39 UTM on 4 May 2017. Because from the analysis of the 45 days of the IOP (Intensive operation period), this was a period during which stationarity conditions could be assumed. The reference (Carvalho, 2019), supporting our statement, is publicly available and the data was plotted in a way that shows the deviation from stationarity. The text was rewritten and information on stratification was also provided.

[Figure]

Figure 1: Bulk Richardson number

L246-250: Is it justifiable to use uniform roughness for this simulation? Have you performed any sensitivity analyses if the sufficient resolution threshold changes when non-uniform roughness is used?

REPLY:

It is justifiable only to reduce the number of variables influencing the problem. Our main objective was to assess the resolution required for the digital terrain model. There was no sensitivity analysis on the effect of non-uniform roughness; this will be the subject of future publications, as stated in the paper. The land coverage at Perdigão is a heterogeneous distribution of eucalyptus and pine trees and will require an approach different from simple characteristic roughness.

L250: Does the impact of the first node height show any convergence, like it is the case for the horizontal resolution? You state that this is worthy of further studies, whereas this is the very study where this should be evaluated.

REPLY:

The impact of the first node height would bring to the discussion other aspects of

computational modelling, beyond the scope of the present paper, focused strictly on the digital terrain model and how accurate the description of the terrain surface must be. A large proportion of Perdigão area is covered by trees, a canopy model will be required to resolve these regions and the first node height is not the only parameter to take into account when using canopy models Lopes da Costa et al. (2006) Silva Lopes et al. (2013).

L278, Figures 12,13: It is not clear what in these figures exactly illustrates the impact of mesh resolution. Are we supposed to see the differences in the insets? Please provide more information in the captions and some discussion about the figures in the text.

    REPLY:
    Figures 12 and 13 were redrawn and the lines in the insets were corrected.

L282: The statement how too high $z_0$ and $u_*$ yield a high loss of momentum is probably badly formulated.

    REPLY:
    The text was rewritten.

L291, Figures 14, 15, 16: There are several revealing features in the figures, but only the absolute wind speed, and to some extent the TKE, receive any discussion. The diverging wind speed in the higher altitudes in Figure 15a is particularly interesting. The legend is hard to read.

    REPLY:
    Figures 14, 15 and 16 were redrawn and the text rewritten.

L294: Please clarify what is meant by "... where a pattern similar to the slope (Table 4) can be observed."

    REPLY:
    The text was rewritten.

L294: Is the RMSE calculated over all the points in the vertical where the measurements are available? The relevance of this statistical measure in a situation where the basic characteristics of the two datasets disagree quite fundamentally (e.g. the TKE at the tower 20/tse04, Figure 14) is probably questionable.

    REPLY:
    No, the RMSE is calculated over the whole profile; it is the RMSE measured against the results on the 20 m resolution mesh (ALS). It is questionable, but once we use this indicator for one variable it should be used for all others.

L304: This conclusion 1. is very categorical, while not entirely based on presented facts. At the resolution of 40 m, which is suggested as sufficient for Perdigão, the RMSE for the SRTM-based simulation is not systematically larger than of the other two datasets; in fact, it can be seen that in terms of RMSE for the 40 m resolution the SRTM-based simulations often have the lowest RMSE (Tables 6, 7, 8).

> REPLY:
> This conclusion is based on the SRTM ability to replicate the ALS data. It is not based on the comparison of flow results using computational meshes from different DTM sources. This conclusion is based on section 4, namely Figures 6 to 11, and Tables 3 and 4, and is based on the ability of the SRTM data to replicate the terrain elevation and slope as measured by the airborne laser scanning (2015).
> Please note that the conclusions are organized into three groups: related to the DTM sources (numbered from 1 to 5), flow model (numbered from 1 to 3) and the final recommendations (numbered from 1 to 3).

**References**

J. P. D. B. Carvalho. Stationarity periods during the Perdigão campaign. Master's thesis, Faculty of Engineering of the University of Porto, 2019. URL `https://repositorio-aberto.up.pt/bitstream/10216/122036/2/348346.pdf`.

J. C. Lopes da Costa, F.A. Castro, J.M.L.M. Palma, and P. Stuart. Computer simulation of atmospheric flows over real forests for wind energy resource evaluation. *Journal of Wind Engineering and Industrial Aerodynamics*, 94:603–620, 2006. doi: 10.1016/j.jweia.2006.02.002. URL `https://www.sciencedirect.com/science/article/pii/S0167610506000328`.

C. Mallet and F. Bretar. Full-waveform topographic lidar: State-of-the-art. *ISPRS Journal of Photogrammetry and Remote Sensing*, 64(1):1–16, January 2009. ISSN 0924-2716. doi: 10.1016/j.isprsjprs.2008.09.007. URL `http://www.sciencedirect.com/science/article/pii/S0924271608000993`.

A. Silva Lopes, J.M.L.M. Palma, and J. Viana Lopes. Improving a two-equation turbulence model for canopy flows using large-eddy simulation. *Boundary-Layer Meteorology*, 149(2): 231–257, November 2013. ISSN 0006-8314, 1573-1472. doi: 10.1007/s10546-013-9850-x. URL `http://link.springer.com/article/10.1007/s10546-013-9850-x`.

---

## Author Response (AR2)

**The digital terrain model in the computational modelling of the flow over the Perdigão site: the appropriate grid size**

authored by

José M. L. M. Palma, Carlos A. M. Silva, Vitor M. C. Gomes, Alexandre Silva Lopes, Teresa Simões, Paula Costa, and Vasco T. P. Batista
Submitted for publication in the *Wind Energy Science*
https://doi.org/10.5194/wes-2019-96

**Reply to the Associate Editor (Jakob Mann)**

Dear Jakob

Please find herewith, our response to your review. After a box, with your text, we address every comment, suggestion or request (either unnumbered or numbered) in two separate sections. I hope that our response is satisfactory and the manuscript can proceed for publication.

Regards
Jose
* * *
**Associate Editor Decision: Publish subject to minor revisions (review by editor)** (22 Jun 2020) by Jakob Mann
Comments to the Author:
The reviews of the paper are quite critical and you chose not to comply with many of the comments. However, I think you need to be more clear in the conclusion about the limitations as both the reviewer request. It should be very clear, also in the conclusion, that the "threshold resolution" is only valid for this particular terrain. It should also be stated that this is not a critical investigation of flow result but that the flow solver is only used to estimate the impact of terrain resolution on the flow. The unrealism of the roughness should also be mentioned there. Please include these matters in the conclusion. You may then call the section "Discussion and conclusion". It is strange that most of the correction in the track changes document are not really changes like "flow" -> "flow" many places in the abstract and so in the rest of the paper. Did something go wrong?
Apart from these changes I would like you to
1) change figure 1: Remove strange box to the left in a). Make the height legend clearer (and with less significant digits. For b) explain the red dots in the figure caption.
2) Figure 11: This is three almost identical figures?!
3) l 297 bad reference "Table ??" and l 62 bad ref to citation.
4) refer to report with more details on the data and where they can be downloaded.
I think the paper is an important and unique contribution, but a bit more care in including the essence of the reviewers criticism is needed and also a bit more care in the preparation of the manuscript.

**1 Unnumbered comments or requests**

- The reviews of the paper are quite critical and you chose not to comply with many of the comments.

  REPLY:
  In our opinion, the criticism of both reviewers is entirely understandable and expectable. Our work is not conventional in the context of neither wind energy nor flow modelling, although being relevant in both areas of application. For instance, many of the references in section 1.1 (Literature survey) are related to geomorphology, aerial surveying or digital terrain modelling, in line with the title. When answering the reviewers, we intended to make this point and, of course, to comply with all their comments.

- However, I think you need to be more clear in the conclusion about the limitations as both the reviewer request. It should be very clear, also in the conclusion, that the "threshold resolution" is only valid for this particular terrain.

  REPLY:
  Because it is not the "threshold resolution" only, the conclusions end with the following sentence:

  > The conclusions hold under the conditions of the present work, namely terrain data and flow model equations and conditions. Under different conditions, further validation may be required.

- It should also be stated that this is not a critical investigation of flow result but that the flow solver is only used to estimate the impact of terrain resolution on the flow.

  REPLY:
  That note is already at the introduction of Section 5 on Flow modelling:

  > Because computational results do not consider, for instance, surface cover heterogeneity, discrepancies are expected when compared with experimental data, which are included here for guidance only.

- The unrealism of the roughness should also be mentioned there. Please include these matters in the conclusion.

  REPLY:
  A new text was added at the end of section 5.1.1

  > However, this [other option than uniform roughness] would increase the number of variables influencing the flow results, masking the effects of the digital terrain model alone. See for instance, the effects of forest resolution and wind orientation relative to the forest stands in the computational modelling of flow over forests in Lopes da Costa et al. (2006).

  The final sentence at the end of the conclusions also covers the case of uniform roughness.

- You may then call the section "Discussion and conclusion".

  REPLY:
  The final section was renamed "Discussion and conclusions", as suggested.

- It is strange that most of the correction in the track changes document are not really changes like "flow" −> "flow" many places in the abstract and so in the rest of the paper. Did something go wrong?

  REPLY:
  The initial version of the manuscript was written with ligatures (`fi`, `fl`, `ff`), as opposed to the latest one, without ligatures (`f\/i`, `f\/l`, `f\/f`). The LaTex package (*latexdiff*) used to identify the differences between `pdf` documents operates on the `.tex` files and identifies these differences hardly noticed in naked eye. In the latest version, the difference is not on the actual word "flow", but on the increased spacing between the `f` and the `l`. The same applies to any word with the letter combinations `fi`, `fl` or `ff`.

**2 Numbered requests**

1) change figure 1: Remove strange box to the left in a). Make the height legend clearer (and with less significant digits. For b) explain the red dots in the figure caption.

   REPLY:
   Figure 1a) was of low quality and unnecessary; because it showed the terrain elevation, repeating information available in other (higher quality) figures, later in the manuscript. Figure 1a) should have been the point density, illustrating the information in the first paragraph. About Figure 1b), explanation on the red dots was included, as requested.

2) Figure 11: This is three almost identical figures?!

   REPLY:
   Yes, they are three almost identical figures. The differences appear at the right end of the horizontal scale; where they should, at the limits of the resolution of either DTM source (SRTM, Mil or ALS). This type of representation is not so uncommon in geomorphology (Nikora and Goring, 2004). Alternative representations, such as histogram would be worse in their ability to show the resolution of the finest terrain details.

   The figures are complex and the interpretation is not easy. Thick lines are the spectra of DTM sources and thin lines the spectra of numerical meshes of different resolution obtained on those sources. The colour code identifies the DTM source. Moving from figures a) to c) (DTM of increasing order of resolution) the numerical meshes (80, 40, 20 and 10 m) are closer to the thick green line; i.e., tend to follow the highest resolution map (ALS). Because, in some cases, the thin and thick lines overlap, the end of every thin line is signalled with

an arrow. In the discussion of these figures, we tried to call the reader's attention to those small and important differences between the three sub-figures.

Figure 11 was redrawn to improve its readability.

3) l 297 bad reference "Table ??" and l 62 bad ref to citation.

REPLY:
You are referring to the file named `Diff` (the outcome of the LaTex package *latexdiff*), with the differences between the initial and the revised versions of our manuscript. The file `wes-2019-96-manuscript-version3.pdf` (revised version), uploaded on 15 Apr 2020, was the one to look at, and there you can see that the reference to Table (Table 6) in line 286 (equivalent to line 297, in file `Diff`) is right[1]. The same applies to line 62; i.e., line 61 in file `wes-2019-96-manuscript-version3.pdf`, where the reference is also correct.

In both cases, we are not sure why that happened. The documentation on *latexdiff* mentions some bugs, which might have been the case. After searching ? on the `Diff` file, we found two more question marks: at lines 76 and 317.

4) refer to report with more details on the data and where they can be downloaded.

REPLY:
The report entitled *Land characterisation of the Perdigão site: data guide* (Palma et al., 2020) is cited at the end of subsection 2.1, beginning of section 4, end of subsection 5.2, and under *Data availability* at the end of the paper.

**New figures:** apart from 1a and 11, Figures 14, 15 and 16 were also redrawn.

[revised manuscript text omitted]